# Functional trade-offs and environmental variation shaped ancient trajectories in the evolution of dim-light vision

Gianni M Castiglione[1,2†], Belinda SW Chang[1,2,3]*

[1]Department of Cell and Systems Biology, University of Toronto, Toronto, Canada; [2]Department of Ecology and Evolutionary Biology, University of Toronto, Toronto, Canada; [3]Centre for the Analysis of Genome Evolution and Function, University of Toronto, Toronto, Canada

**Abstract** Trade-offs between protein stability and activity can restrict access to evolutionary trajectories, but widespread epistasis may facilitate indirect routes to adaptation. This may be enhanced by natural environmental variation, but in multicellular organisms this process is poorly understood. We investigated a paradoxical trajectory taken during the evolution of tetrapod dim-light vision, where in the rod visual pigment rhodopsin, E122 was fixed 350 million years ago, a residue associated with increased active-state (MII) stability but greatly diminished rod photosensitivity. Here, we demonstrate that high MII stability could have likely evolved *without* E122, but instead, selection appears to have entrenched E122 in tetrapods *via* epistatic interactions with nearby coevolving sites. In fishes by contrast, selection may have exploited these epistatic effects to explore alternative trajectories, but *via* indirect routes with low MII stability. Our results suggest that within tetrapods, E122 and high MII stability cannot be sacrificed—not even for improvements to rod photosensitivity.

DOI: https://doi.org/10.7554/eLife.35957.001

*For correspondence:
belinda.chang@utoronto.ca

Present address: [†]Department of Ophthalmology, Johns Hopkins University School of Medicine, Baltimore, United States

**Competing interests:** The authors declare that no competing interests exist.

## Introduction

Nature-inspired strategies are increasingly recruited toward engineering objectives in protein design (*Khersonsky and Fleishman, 2016*; *Jacobs et al., 2016*; *Goldenzweig and Fleishman, 2018*, a central challenge of which is to successfully manipulate backbone structure to modulate stability without introducing undesirable pleiotropic effects on protein activity (*Khersonsky and Fleishman, 2016*; *Goldenzweig and Fleishman, 2018*; *Starr and Thornton, 2017*; *Tokuriki and Tawfik, 2009*). Engineering protein stability and activity requires an understanding of a protein's sequence-function relationship, or landscape (*Pál and Papp, 2017*; *Wu et al., 2016*; *Starr et al., 2017*), where billions of possible pair-wise and third-order interactions can exist between amino acids (*Starr and Thornton, 2017*; *Storz, 2016*), and only a limited number of amino acid combinations will confer the function of interest (*Wu et al., 2016*; *Starr et al., 2017*; *McMurrough et al., 2014*; *Mateu and Fersht, 1999*; *Tarvin et al., 2017*). To understand the context-dependence of amino acid functional effects (also known as intramolecular epistasis [*Starr et al., 2017*; *Storz, 2016*; *Echave et al., 2016*]), approaches such as deep mutational scanning (*Wu et al., 2016*; *Starr et al., 2017*; *Sailer and Harms, 2017*) can explore a subset of sequence-function space formed in response to a limited set of artificial selection pressures (*Starr and Thornton, 2017*). By contrast, natural protein sequence variation reflects the range of protein function that evolved in response to changing ecological variables (*Starr and Thornton, 2017*; *Pál and Papp, 2017*; *Ogbunugafor et al., 2016*), where convergent 'solutions' for protein function and stability can be derived through the evolution of alternative protein sequences (*McMurrough et al., 2014*; *Mateu and Fersht, 1999*; *Tarvin et al., 2017*). This

**eLife digest** People can see in dim light because of cells at the back of the eye known as rods. These cells contain two key components: molecules called retinal, which are bound to proteins called rhodopsin. When light hits a rod cell, it kicks off a cascade of reactions beginning with the retinal molecule changing into an activated shape and ending with a nerve impulse travelling to the brain. The activated form of retinal is toxic, and as long as it remains bound to the rhodopsin protein it will not damage the rod or surrounding cells. The toxic retinal also cannot respond to light. It must be released from the protein and converted back to its original shape to restore dim light vision.

As with all proteins, rhodopsin's structure comprises a chain of building blocks called amino acids. Every land animal with a backbone has the same amino acid at position 122 in its rhodopsin. This amino acid, named E122, helps to stabilize the activated rhodopsin, slowing the release of the toxic retinal. Yet E122 also makes the rod cells less sensitive, resulting in poorer vision in dim light. In contrast, some fish do not have E122 but rather one of several different amino acids takes its place. What remains unclear is why all land animals have stuck with E122, and whether there were other options that evolution could have explored to overcome the trade-off between sensitivity and stability.

By looking at the make-up of rhodopsins from many animals, Castiglione and Chang found other sites in the protein where the amino acid changed whenever position 122 changed. The amino acids at these so-called "coevolving sites" were then swapped into the version of rhodopsin that is found in cows, which had also been engineered to lack E122. These changes fully compensated for the destabilizing loss of E122 on activated rhodopsin but without sacrificing its sensitivity to light. Further experiments then confirmed that unless all amino acids were substituted at once, the activated rhodopsin was very unstable. Indeed, it was almost as unstable as mutated rhodopsins found in some human diseases. These findings suggest that, while there was in principle another solution available to land animals, the routes to it were closed off because they all came with an increased risk of eye disease.

These findings highlight that rhodopsin likely plays a more important role in protecting humans and many other land animals against eye disease than previously assumed. More knowledge about this protective role may lead to new therapies for these conditions. Also, investigating similar evolutionary trade-offs could help to explain how and why different proteins work the way that they do today.

DOI: https://doi.org/10.7554/eLife.35957.002

suggests that closer examination of natural sequence variation may reveal new blueprints for protein design.

The dim-light visual pigment rhodopsin (RH1/RHO) is an excellent model for understanding how both ecological variables and biophysical pleiotropy may interact to determine the availability of functional evolutionary solutions for environmental challenges (*Kojima et al., 2017*; *Gozem et al., 2012*; *Dungan and Chang, 2017*; *Castiglione et al., 2018*). Spectral tuning mutations that shift the RH1 wavelength of maximum absorbance ($\lambda_{MAX}$) can adapt dim-light vision to a remarkable range of spectral conditions across aquatic and terrestrial visual ecologies (*Hunt et al., 2001*; *Hauser and Chang, 2017a*; *Dungan et al., 2016*). Recently, $\lambda_{MAX}$ was revealed to exist within a complex series of epistasis-mediated trade-offs with the non-spectral functional properties of RH1 long understood as adaptations for dim-light (*Gozem et al., 2012*; *Dungan and Chang, 2017*; *Castiglione et al., 2017*; *Hauser et al., 2017b*). These include an elevated barrier to spontaneous thermal-activation, which minimizes rod dark noise and is promoted by blue-shifts in $\lambda_{MAX}$ (*Kojima et al., 2017*; *Gozem et al., 2012*; *Kefalov et al., 2003*; *Yue et al., 2017*); and a slow decay of its light-activated conformation, which we refer to here as metarhodopsin-II (MII) for simplicity (*Imai et al., 1997*; *Lamb et al., 2016*; *Kojima et al., 2014*; *Sommer et al., 2012*; *Schafer et al., 2016*; *Van Eps et al., 2017*). The RH1 MII active conformation is associated with rapid and efficient activation of G-protein transducin ($G_t$) (*Kojima et al., 2014*; *Sugawara et al., 2010*), yet the reasons for its long-decay after $G_t$ signaling remain unclear (*Kefalov et al., 2003*; *Imai et al., 1997*; *Imai et al., 2007*).

To sustain vision, all-*trans* retinal (atRAL) chromophore must be released from MII after $G_t$ signaling (*Palczewski, 2006*)—a process that depends on the conformational stability of the MII-active-state structure (*Schafer et al., 2016*; *Schafer and Farrens, 2015*). Cone opsins have low MII stability and therefore rapidly release atRAL (*Imai et al., 1997*; *Chen et al., 2012a*), where it is quickly recycled back into 11-*cis* retinal (11CR) through the cone visual (retinoid) cycle, enabling rapid regeneration of cone pigments for bright-light vision (*Wang and Kefalov, 2011*; *Tsybovsky and Palczewski, 2015*). Rods, in contrast, regenerate thousands of times slower than cones after bright-light exposure (*Mata et al., 2002*). Indeed, rod exposure to bright flashes of light leads to atRAL release that can outpace clearance by visual cycle enzymes (*Sommer et al., 2014*; *Rózanowska and Sarna, 2005*), thus leading to accumulation (*Saari et al., 1998*; *Lee et al., 2010*) and light-induced retinopathy through various modes of cellular toxicity involving oxidative stress (*Maeda et al., 2009*; *Chen et al., 2012b*). Interestingly, recent biochemical evidence suggests MII may play a role in retinal photoprotection by complexing with arrestin after $G_t$ signaling to re-uptake and thus provide a sink for toxic atRAL after rod photobleaching (*Sommer et al., 2014*). This suggests the evolution of rhodopsin's high conformational selectivity for toxic atRAL may be a functional specialization (*Schafer et al., 2016*; *Schafer and Farrens, 2015*), which could in turn reflect differences in retinoid metabolism between rods vs. cones (*Wang and Kefalov, 2011*; *Tsybovsky and Palczewski, 2015*; *Imai et al., 2005*).

Consistent with the overlapping mechanisms of RH1 spectral and non-spectral functions *via* the highly constrained RH1 structure (*Gozem et al., 2012*; *Yue et al., 2017*), this biophysical pleiotropy likely necessitates costly trade-offs between the spectral and non-spectral functions of RH1 in natural systems (*Dungan and Chang, 2017*; *Luk et al., 2016*). By comparison, directed evolution and synthetic biology approaches have successfully engineered either spectral, or non-spectral aspects of rhodopsin function, but did not address trade-offs arising from shifts in function. It has thus been possible to shift the spectral absorbance of archaea and bacterial rhodopsins close to the limit of the visible spectrum (*Herwig et al., 2017*; *McIsaac et al., 2014*), and to engineer tetrapod rhodopsins with high thermal stability (*Xie et al., 2003*), constitutive activation (*Deupi et al., 2012*; *Standfuss et al., 2011*), and alternative chromophore-binding sites (*Devine et al., 2013*). However, it has not been investigated whether rod visual pigments with novel combinations of spectral *and* non-spectral functional properties can be engineered by manipulating the biophysical pleiotropy of RH1 otherwise exploited by natural selection.

Site 122 (*Bos taurus* RH1 numbering) is a molecular determinant of both the spectral and non-spectral functional properties of rhodopsin and the cone opsins (*Hunt et al., 2001*; *Yue et al., 2017*; *Imai et al., 1997*; *Imai et al., 2007*; *Yokoyama et al., 1999*). Intriguingly, vertebrate visual pigment families show differences in which amino acid variants predominate at this site (*Figure 1A*), with I122 strongly conserved in the most ancestrally diverging cone opsins such as the long-wave sensitive opsins (LWS) (*Lamb et al., 2007*), whereas in the most derived opsin group, the rhodopsins (RH1), E122 predominates (*Figure 1B,C*) (*Imai et al., 1997*; *Lamb et al., 2007*; *Imai et al., 2007*; *Carleton et al., 2005*). E122 is a key component of an important hydrogen-bonding network with H211 that is known to stabilize the MII active-conformation (*Choe et al., 2011*). This stability increase is so dramatic that E122 is considered a functional determinant distinguishing rhodopsin from cone opsins (*Figure 1B*) (*Imai et al., 1997*; *Lamb et al., 2016*; *Kojima et al., 2014*). Paradoxically, by conferring this increase in MII stability, the evolution of E122 likely involved a costly fitness trade-off that diminished tetrapod rod photosensitivity (*Yue et al., 2017*), which can affect visual performance in animals (*Kojima et al., 2017*; *Aho et al., 1988*). Indeed, it is possible to improve tetrapod rod photoreceptor sensitivity by decreasing rod dark noise in vivo by replacing E122 with a cone opsin amino acid variant (COV; *Figure 1A*) at site 122, such as Q122, which predominates in RH2 cone opsins (*Yue et al., 2017*; *Lin et al., 2017*). The strict conservation of E122 in all tetrapod rhodopsins (*Figure 1C*, *Table 1*, *Supplementary file 1*) therefore suggests that during the evolution of tetrapod dim-light vision, natural selection may have prioritized MII stability (*Figure 1B,D*) at the *expense* of rod sensitivity. This apparent evolutionary trade-off is perplexing given that the low spontaneous thermal activation of rhodopsin (and therefore rod dark noise) is a functional hallmark of rhodopsin divergence from the cone opsins (*Kojima et al., 2017*; *Gozem et al., 2012*; *Kefalov et al., 2003*; *Lamb et al., 2016*).

Why has tetrapod RH1 been constrained to this paradoxical compromise at site 122 for the last 350 million years? Interestingly, and in contrast to tetrapod rhodopsins, fish rhodopsins show

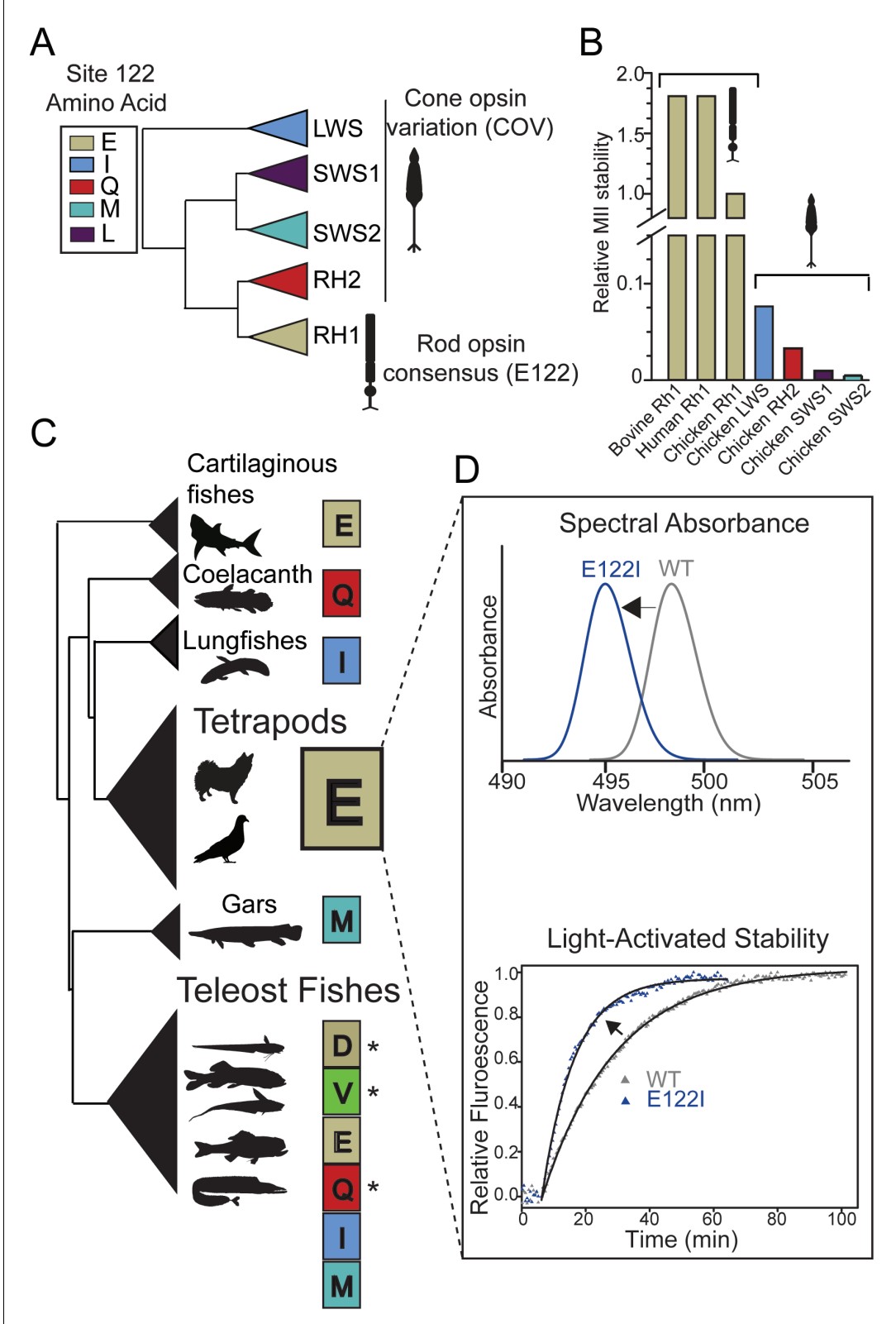

**Figure 1.** Natural variation at site 122 determines rhodopsin function and stability. (**A**) Amino acid consensus residues at site 122 across vertebrate rod opsins (rhodopsin; RH1) and the cone opsins (long-wave (LWS), short-wave (SWS1 and SWS2) and middle-wave (RH2) sensitive). Modified from (*Lamb et al., 2007*). (**B**) Relative stability of the rod and cone opsin active-conformation (MII) in different vertebrates (*Imai et al., 2005*). (**C**) Schematic representation of naturally occurring cone opsin variants (COVs) and other amino acids across vertebrate RH1 (see *Figure 1—figure supplements 1–2*; *Figure 1 continued on next page*

*Figure 1 continued*

*Tables 1–2, Supplementary files 1–2*). E122 is invariant in all Tetrapod RH1 genes sequenced to date. Natural deep-sea amino acid variants (*Hunt et al., 2001*; *Yokoyama et al., 1999*) are identified with an asterisk (\*; *Table 2*). (**D**) Introduction of the ancestral cone opsin (LWS) variant I122 blue shifts tetrapod RH1 spectral absorbance and accelerates decay of the MII light-activated conformation.

DOI: https://doi.org/10.7554/eLife.35957.003

The following figure supplements are available for figure 1:

**Figure supplement 1.** Schematic of RH1 site 122 variation across the vertebrate phylogeny.

DOI: https://doi.org/10.7554/eLife.35957.004

**Figure supplement 2.** Vertebrate phylogeny used in computational analyses.

DOI: https://doi.org/10.7554/eLife.35957.005

variation at site 122, such as in the Coelacanth (*Latimeria chalumnae*), Lungfish (*Neoceratodus forsteri*), and deep-sea fish lineages, where COV (I, Q, M) and other residues at site 122 (V, D) are found (*Figure 1C*; *Figure 1—figure supplement 1*; *Tables 1–2, Supplementary file 2*) (*Hunt et al., 2001*; *Yokoyama et al., 1999*; *Carleton et al., 2005*). These substitutions have been shown to blue-shift $\lambda_{MAX}$ by up to ~10 nm (*Hunt et al., 2001*; *Yokoyama et al., 1999*), and may improve dim-light sensitivity in poorly-lit aquatic environments (*Yue et al., 2017*). Strikingly, one of the largest freshwater groups—the Characiphysi (which includes piranhas, electric eels, and catfishes [*Chen et al., 2013*]) —has the COV I122 residue completely fixed (*Figure 1—figure supplement 1*, *Supplementary file 3*). In tetrapods by contrast, the red-shifting E122 mutation is strictly maintained, increasing MII stability (*Imai et al., 1997*) but greatly decreasing rod sensitivity (*Yue et al., 2017*). Why the strong constraints on high MII stability and E122 are relaxed only within certain aquatic visual ecologies, remains unknown.

In light of these ecological patterns, we questioned whether it was possible to synthesize an evolutionary alternative: a tetrapod RH1 that never lost COV at site 122 but still developed high MII stability. We reasoned that relative to tetrapods, the diversity and complexity of fish visual ecologies

**Table 1.** Variation at sites 119-122-123-124 in Tetrapods and Outgroup rh1.

Sites with variation relative to the Vertebrate consensus (LEIA) are in bold and highlighted grey. Subterranean species are denoted (\*).

| | Species | Accession | Common name | 119 | 122 | 123 | 124 |
|---|---|---|---|---|---|---|---|
| Outgroups | *Callorhinchus milii* | XP_007888679 | Elephant shark | L | E | I | **G** |
| | *Orectolobus ornatus* | AFS63882 | Ornate wobbegong | L | E | **V** | **S** |
| | *Latimeria chalumnae* | XP_005997879 | Coelacanth | L | **Q** | **V** | A |
| | *Neoceratodus forsteri* | ABS89278 | Australian lungfish | **F** | **I** | I | A |
| Mammals | *Dasypus novemcinctus* | XP_004477303 | 9-banded armadillo\* | **I** | E | I | A |
| | *Eptesicus fuscus* | XP_008150514 | Big brown bat | L | E | **V** | A |
| | *Chrysochloris asiatica* | XP_006868732 | Cape golden mole\* | **M** | E | I | A |
| | *Sorex araneus* | XP_004613289 | Common Shrew\* | L | E | **V** | A |
| | *Tupaia chinensis* | XP_006160726 | Tree Shrew | L | E | **V** | A |
| | *Ictidomys tridecemlineatus* | XP_005333841 | 13-line ground squirrel | L | E | **V** | A |
| | *Rattus norvegicus* | NP_254276 | brown rat | L | E | I | **G** |
| | *Sarcophilus harrisii* | XP_003762497 | Tasmanian devil | **T** | E | **V** | A |
| Reptiles | *Alligator mississippiensis* | XM_006274155 | American alligator | L | E | **V** | A |
| | *Alligator sinensis* | XP_006039462 | Chinese alligator | L | E | **V** | A |
| | *Anolis carolinensis* | NP_001278316 | Carolina anole | L | E | **M** | **G** |
| | *Python bivittatus* | XP_007423324 | Burmese python | L | E | **M** | A |
| Amphibians | *Ambystoma tigrinum* | U36574 | Tiger salamander | **M** | E | I | A |
| | *Cynops pyrrhogaster* | BAB55452 | Jap. Fire belly newt | L | E | I | **G** |
| | *Xenopus tropicalis* | NP_001090803 | Western clawed frog | L | E | **M** | A |
| | *Xenopus laevis* | NP_001080517 | African clawed frog | L | E | **V** | A |

DOI: https://doi.org/10.7554/eLife.35957.006

**Table 2.** Fish rh1 with variation at site 122 do not necessarily have variation at coevolving sites 119, 123, and 124. Sites with variation relative to the Vertebrate consensus (LEIA) are in bold and highlighted grey.

| Order | Species | Accession | Common name | 119 | 122 | 123 | 124 | Ecology notes from FishBase |
|---|---|---|---|---|---|---|---|---|
| Lepisosteiformes | *Lepisosteus oculatus* | JN230969.1 | spotted gar | L | **M** | I | **S** | Freshwater; brackish; demersal. (Ref. 2060) |
| | *Atractosteus tropicus* | JN230970.1 | Tropical Gar | L | **M** | **L** | **S** | Freshwater; demersal |
| Osteoglossiformes | *Mormyrops anguilloides* | JN230973.1 | Cornish Jack | **T** | **I** | I | A | Freshwater; demersal; potamodromous (Ref. 51243) |
| | *Osteoglossum bicirrhosum* | KY026030.1 | Silver arowana | **T** | **I** | I | A | Freshwater; benthopelagic |
| Alepocephalifromes | *Alepocephalus bicolor* | JN230974.1 | Bicolor slickhead | L | **Q** | I | A | Marine; bathydemersal; depth range 439–1080 m (Ref. 44023). |
| | *Bathytroctes microlepis* | JN544540.1 | Smallscale smooth-head | L | **D** | I | A | Marine; bathypelagic; depth range 0–4900 m (Ref. 58018) |
| | *Conocara salmoneum* | JN412577.1 | Salmon smooth-head | L | **Q** | I | A | Marine; bathypelagic; depth range 2400–4500 m (Ref. 40643) |
| Galaxiiformes | *Galaxias maculatus* | JN231000.1 | Inanga | L | **M** | I | **G** | Marine; freshwater; brackish; benthopelagic; catadromous (Ref. 51243). |
| Stomiatiformes | *Argyropelecus aculeatus* | JN412571.1 | Lovely Hatchetfish | **H** | **Q** | I | A | Marine; bathypelagic; depth range 100–2056 m (Ref. 27311) |
| | *Vinciguerria nimbaria* | JN412570.1 | Oceanic lightfish | **H** | **Q** | **V** | A | Marine; bathypelagic; depth range 20–5000 m (Ref. 4470) |
| Ateleopodiformes | *Ateleopus japonicus* | KC442218.1 | Pacific Jellynose Fish | L | **M** | I | **S** | Marine; bathydemersal; depth range 140–600 m (Ref. 44036). |
| Myctophiformes | *Benthosema suborbitale* | JN412576.1 | Smallfin lanternfish | **H** | **Q** | **V** | **G** | Marine; bathypelagic; oceanodromous; depth range 50–2500 m (Ref. 26165) |
| | *Lampanyctus alatus* | JN412575.1 | Winged lanternfish | **H** | **Q** | **V** | A | Marine; bathypelagic; oceanodromous; depth range 40–1500 m (Ref. 26165) |
| | *Neoscopelus microchir* | KC442224.1 | Shortfin neoscopelid | L | **Q** | I | A | Marine; bathypelagic; depth range 250–700 m (Ref. 4481) |
| Gadiiformes | *Coryphaenoides guentheri* | JN412578.1 | Gunther's grenadier | L | **V** | I | A | Marine; bathydemersal; depth range 831–2830 (Ref. 1371) |
| Beryciformes | *Melamphaes suborbitalis* | JN231006.1 | Shoulderspine bigscale | L | **Q** | I | A | Marine; brackish; bathypelagic; depth range 500–1000 m (Ref. 31511). |
| Holocentriformes | *Holocentrus rufus* | KC442230.1 | Longspine squirrelfish | L | **M** | I | **S** | Marine; reef-associated; depth range 0–32 m (Ref. 3724). |
| | *Myripristis murdjan* | KC442231.1 | Pinecone soldierfish | L | **M** | I | **G** | Marine; reef-associated; depth range 1–50 m (Ref. 9710) |
| Scombriformes | *Aphanopus carbo* | EU637938.1 | Black scabbardfish | **H** | **Q** | I | **G** | Marine; bathypelagic; oceanodromous (Ref 108735); 200–2300 m (Ref. 108733) |
| | *Cubiceps gracilis* | EU637952.1 | Driftfish | - | **Q** | I | A | Marine; pelagic-oceanic; oceanodromous (Ref. 51243); |

DOI: https://doi.org/10.7554/eLife.35957.007

(*Hunt et al., 2001*; *Hauser and Chang, 2017a*) may have allowed selection the opportunity to explore the pleiotropic potential of site 122 through the evolution of novel structural interactions with nearby sites that could compensate for the destabilizing loss of the E122-H211 hydrogen bond. To identify these interactions, our goal was to use analyses of evolutionary rates to predict sites coevolving with site 122, and to investigate the functional consequences of coevolving sites with experimental site-directed mutagenesis studies. Ultimately, we used our analyses of natural variation as a guide to artificially engineer a tetrapod rhodopsin with increased MII stability, but within a non-E122 sequence background. We demonstrated that this synthetic alternative is possible, even if evolution did not proceed down this mechanistic trajectory toward a dim-light adapted visual pigment.

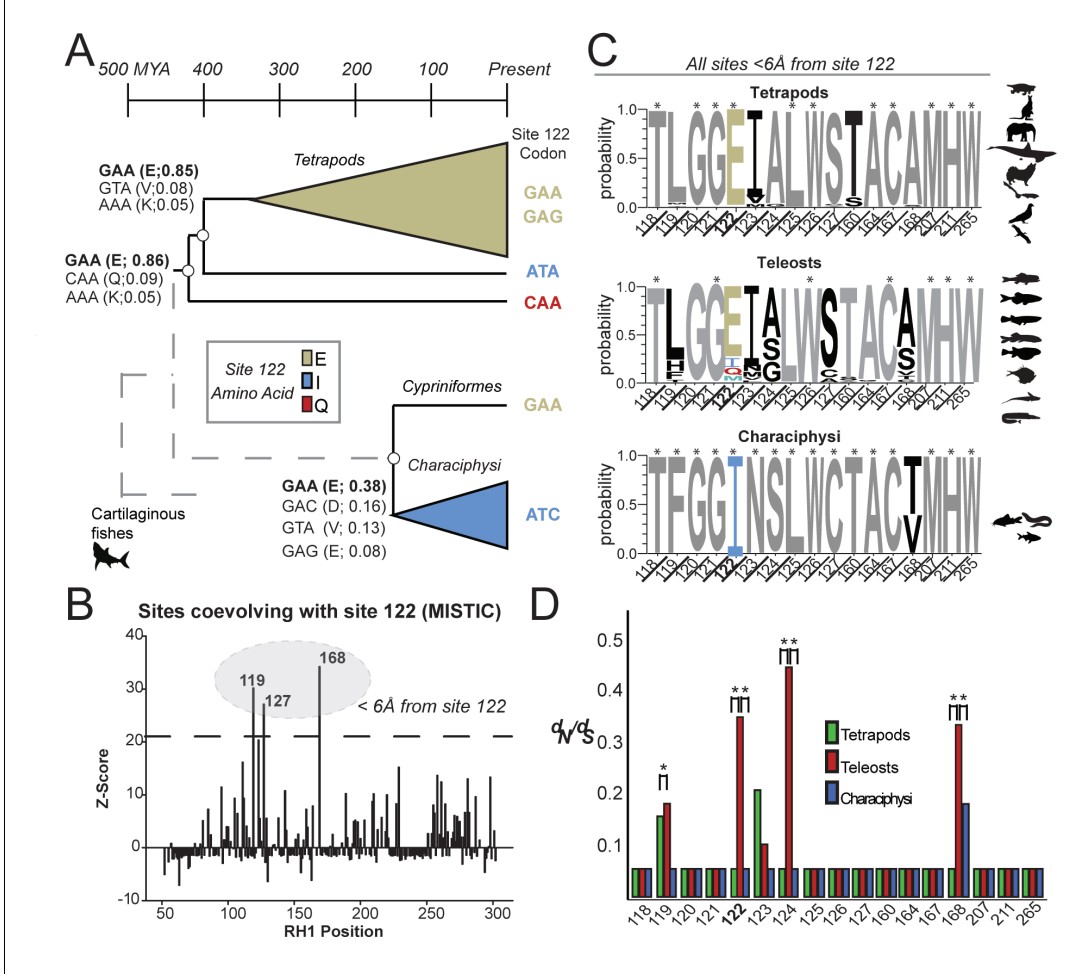

**Figure 2.** Local coevolutionary forces govern the evolution of site 122 differentially between tetrapods and fish (teleost) RH1. (**A**) Extant and reconstructed codon variation at site 122 (Materials and methods). Despite a variety of residues at site 122 across the Coelacanth (Q122), Lungfish (I122; Ceratodontiformes), and Tetrapods (E122), GAA codons encoding for E122 are nevertheless predicted as the ancestral state with high posterior probabilities (shown in parentheses). E122 (GAA/GAG) is also likely to have been present in the last common ancestor of Cypriniformes and the Characiphysi, although with low posterior probabilities and therefore high uncertainty. I122 codon ATC is fixed in all Characiphysi rhodopsin to our knowledge (*Supplementary file 3*). Approximate divergence times are from (*Hedges et al., 2015*). (**B**) Mutual information (MI) analyses (MISTIC [*Simonetti et al., 2013*]) reveal all sites coevolving with site 122 are within 6 Å. Significance thresholds were determined by reference to the highest MI z-score from all sites across analyses of randomized datasets (n = 150; z-score cut-off = 21.6), as previously described (*Ashenberg and Laub, 2013*). (**C**) Sites within this radius displayed decreased amino acid variation in tetrapod and characiphysi RH1, where E122 and I122 are fixed, respectively (asterisks). (**D**) In tetrapods and characiphysi RH1, reduction in amino acid variation (relative to teleosts) at positions within the 6 Å radius were driven by increases in purifying selection on non-synonymous codons. Statistically significant gene-wide increases in purifying selection (*) between lineages were detected by likelihood ratio tests of alternative (Clade model C [*Bielawski and Yang, 2004*]) and null (M2a_REL [*Weadick and Chang, 2012*]) model analyses of codon substitution rates ($d_N/d_S$) ((p<0.001); *Tables 3–5*). Sites estimated to be under this increase in purifying selection (*) were those identified in the divergent site class of the CmC model analyses through a Bayes empirical Bayes analysis as previously described (*Castiglione et al., 2017*. Site-specific $d_N/d_S$ estimates are from M8 analyses on phylogenetically pruned datasets (*Tables 8–10*; *Figure 1—figure supplement 2*; *Figure 2—figure supplement 1*).

DOI: https://doi.org/10.7554/eLife.35957.008

The following figure supplement is available for figure 2:

**Figure supplement 1.** Phylogeny used in PAML computational analyses of Characiphysi rhodopsin coding-sequences, along with outgroups (*Table 10*; *Supplementary file 3*).

DOI: https://doi.org/10.7554/eLife.35957.009

**Table 3.** Results of Clade Model C (CmC) analyses of vertebrate *rh1* under various partitions.

| Model and Foreground[†] | ΔAIC[‡] | *ln*L | Parameters | | | Null | *P* [df] |
|---|---|---|---|---|---|---|---|
| | | | $\omega_0$ | $\omega_1$ | $\omega_2/\omega_d$ | | |
| M2a_rel | 225.5 | −47185.37 | 0.02 (69%) | 1 (3%) | 0.20 (28%) | N/A | - |
| CmC_*Tetrapod Branch* | 97.44 | −47119.33 | 0.20 (28%) | 1 (3%) | 0.02 (69%) Tetra Br: 0.00 | M2a_rel | 0.000 [1] |
| CmC_*Tetrapod* | 4.92 | −47073.06 | 0.02 (67%) | 1 (3%) | 0.24 (30%) Tetra: 0.13 | M2a_rel | 0.000 [1] |
| CmC_*Teleost* | 1.88 | −47071.54 | 0.02 (67%) | 1 (3%) | 0.14 (30%) Teleost: 0.24 | M2a_rel | 0.000 [1] |
| CmC_*Teleost vs Tetrapod* | 0* | −47069.60 | 0.02 (67%) | 1 (3%) | 0.17 (30%) Tetra: 0.13 Teleost: 0.24 | M2a_rel | 0.000 [2] |

[†]The foreground partition is listed after the underscore for the clade models and consists of either: the clade of Teleost fishes (*Teleost*); the clade Tetrapods (*Tetrapod;Tetra*) or branch leading to tetrapods (*Tetrapod branch; Tetra Br*); or the clades of both the teleost fishes and tetrapods as two separate foregrounds (*Teleost vs Tetrapods*). In any partitioning scheme, the entire clade was tested, and all non-foreground data are present in the background partition.

[‡]All ΔAIC values are calculated from the lowest AIC model. The best fit is shown with an asterisk (*).

$\omega_d$ is the divergent site class, which has a separate value for the foreground and background partitions.

[¶]Significant p-values (α ≤0.05) are bolded. Degrees of freedom are given in square brackets after the p-values.

Abbreviations—*ln*L, ln Likelihood; p, p-value; AIC, Akaike information criterion.

DOI: https://doi.org/10.7554/eLife.35957.010

# Results

## Phylogenetic identification of an intramolecular coevolutionary network

To better understand the selection pressures that may be constraining E122 to fixation during tetrapod evolution, we constructed a large vertebrate rhodopsin phylogenetic dataset (*Figure 1—figure supplements 1* and *2*, *Supplementary files 1–2*) and investigated the evolutionary history of site 122 using ancestral reconstruction (Materials and methods). We found that E122 (codon GAA; *Figure 2A*) has been fixed in tetrapod RH1 since the most recent common ancestor ~350 million years ago (MYA) (*Hedges et al., 2015*), where it appears along the ancestral branch leading to tetrapods (*Figure 2A*; *Table 3*) following the diversification from lungfishes (I122, codon ATA, *Figure 2A*; *Supplementary file 1*) and the coelacanth (Q122, codon CAA, *Figure 2A*; *Supplementary file 1*). This transition period in vertebrate evolution is characterized by extensive morphological modifications for vision within terrestrial environments, and likely included large increases in environmental light irradiance (*MacIver et al., 2017*; *Warrant and Johnsen, 2013*). Apart from the lungfishes and coelacanth, the high conservation of E122 in tetrapods is also reflected in other vertebrate rhodopsins (*Figure 1*, *Figure 1—figure supplement 1*; *Tables 1–2*; *Supplementary files 1–2*), but there are important exceptions within certain lineages of teleost fishes, such as the Characiphysi. Within this group, the COV residue I122 was introduced likely through E122I (codon ATC; *Figure 2A*), where I122 is now completely fixed across the extant Characiphysi (*Supplementary file 3*).

Since fishes (Teleosts), unlike tetrapods, display amino acid variation at site 122 (*Figure 1C*), we hypothesized that compensatory mutations may be coevolving with site 122 across fish RH1. To test this hypothesis, we investigated across the entire transmembrane domain of rhodopsin (residues 53–302) for evidence of sites coevolving with site 122 within an alignment of Teleost RH1 (Materials and methods; *Supplementary file 2*). Using phylogenetically corrected mutual information (MI) analyses (MISTIC; [(*Simonetti et al., 2013*)]) with z-score cut-off determined by analyses of randomized datasets (*Ashenberg and Laub, 2013*), we found significant evidence of coevolution with site 122 at several RH1 positions, all of which clustered within 6 Å of E122 (*Figure 2B*) in the MII crystal structure (*Choe et al., 2011*. This is within the range at which intramolecular forces such as Van der Waals and hydrophobic interactions between amino acids are thought to occur (*Ivankov et al., 2014*). It is known, however, that there is a tendency of covariation analyses such as MI to identify coevolving sites proximal to each other, which may in turn overlook more distal coevolving sites potentially

**Table 4.** Analyses used to elucidate sites coevolving with site 122 in Vertebrate rhodopsins (*rh1*).

In bold are the results of interest described in the main text, including: elevated $d_N/d_S$, long-term shifts in selection between teleosts and tetrapods, amino acid statistical covariation with site 122 in the teleost dataset, and phylogenetically correlated amino acid variation with site 122.

| Site | Distance to site 122 (Å)[*] | Tetrapod M8 $d_N/d_S$[†] | Teleost M8 $d_N/d_S$[†] | Characiphysi M8 $d_N/d_S$[†] | Posterior probability of long-term shift in selection (tetrapod/characiphysi)[‡] | Z-score covariation[§] | Significant correlated evolution? |
|---|---|---|---|---|---|---|---|
| 118 | 5 | 0.05 | 0.05 | 0.05 | 0.00/0.00 | −1.54 | No |
| 119 | 3.5 | **0.14** | **0.168** | **0.05** | 0.57/0.19 | **30.2** | **Yes** |
| 120 | 3.1 | 0.05 | 0.05 | 0.05 | 0.00/0.00 | −1.91 | No |
| 121 | N/A | 0.05 | 0.05 | 0.05 | 0.00/0.00 | −0.94 | No |
| 122 | N/A | **0.05** | **0.322** | **0.05** | 1.00/1.00 | N/A | N/A |
| 123 | N/A | 0.19 | 0.094 | 0.05 | 0.00/0.00 | 20.4 | No |
| 124 | 3.2 | **0.05** | **0.411** | **0.05** | 1.00/1.00 | 5.53 | No |
| 125 | 3.4 | 0.05 | 0.05 | 0.05 | 0.00/0.00 | 1.19 | No |
| 126 | 3.7 | 0.05 | 0.05 | 0.05 | 0.00/0.00 | −1.60 | No |
| 127 | 4.9 | 0.05 | 0.065 | 0.05 | 0.00/0.00 | **27.1** | **Yes** |
| 160 | 5.1 | 0.05 | 0.05 | 0.05 | 0.00/0.00 | 3.90 | No |
| 164 | 4.2 | 0.05 | 0.05 | 0.05 | 0.00/0.00 | 7.88 | No |
| 167 | 3.8 | 0.05 | 0.05 | 0.05 | 0.00/0.00 | −1.38 | No |
| 168 | 5.9 | **0.05** | **0.308** | **0.192** | 1.00/1.00 | 34.2 | No |
| 207 | 4.9 | 0.05 | 0.05 | 0.05 | 0.00/0.00 | −1.21 | No |
| 211 | 2.7 | 0.05 | 0.05 | 0.05 | 0.00/0.00 | −1.38 | No |
| 265 | 5.1 | 0.05 | 0.05 | 0.05 | 0.00/0.00 | −1.44 | No |

[*]From structural analysis of distances between amino acids and site 122 within the MII crystal structure 3PQR (**Choe et al., 2011**).

[†]Post mean $d_N/d_S$ from M8 analyses described in **Tables 8–10**.

[‡]Bayes empirical Bayes posterior probability of long-term shift in selection calculated in Clade model C (CmC) (**Yang, 2007**) analyses (CmC_*Teleost* vs *Tetrapod*/CmC_*Characi* clade) described in **Tables 3** and **5**, respectively.

[§]Phylogenetically corrected MI z-scores (MISTIC; [**Simonetti et al., 2013**]) of covariation with site 122 from analyses on Teleost RH1 dataset. Values were considered significant if greater than the top absolute z-score (21.6) from all site-wise comparisons from all analyses of 150 randomized datasets, as described (**Ashenberg and Laub, 2013**).

[¶]Tests of correlated evolution in amino acid variation (**Pagel, 1994**) between a given site and site 122. p-values were calculated by performing Monte Carlo tests using data from simulations (n > 1000) in MESQUITE (**Maddison and Maddison, 2017**). p-Values were subjected to a Bonferroni-correction to determine significance (p<0.002).

DOI: https://doi.org/10.7554/eLife.35957.011

indirectly interacting with site 122 (*Talavera et al., 2015*). Nevertheless, sites detected within this 6 Å radius (sites 119, 123) have been previously found capable of functionally compensating for human pathogenic mutations (e.g. A164V) disrupting the MII-stabilizing E122-H211 interaction (*Stojanovic et al., 2003*), suggesting that natural variation at coevolving sites within this radius could compensate for the functional effects of COV at site 122.

We therefore decided to focus our investigations on identifying natural compensatory mutations at sites within this 6 Å radius. Relative to Teleost RH1 (where site 122 varies), we found that sites within this radius displayed decreased amino acid variation in Tetrapod and Characiphysi RH1, where E122 and I122 are fixed, respectively (asterisks, *Figure 2C*). This observation is consistent with an intramolecular evolutionary process known as entrenchment (*Pollock et al., 2012*; *Goldstein and Pollock, 2017*; *Shah et al., 2015*), where functionally favourable amino acid residues compensating for an original mutation tend to become fixed, thus mutually entrenching favourable amino acids at each position within the coevolving network. We therefore reasoned that if residues at nearby positions are indeed compensatory, then these sites should display a relative decrease in amino acid variation specifically in those vertebrate lineages where an amino acid has been fixed at site 122– such as E122 in tetrapods and I122 in the Characiphysi. Furthermore, we hypothesized that decreases in

**Table 5.** Results of Clade Model C (CmC) analyses of teleost *rh1* under various partitions.

| Model and Foreground[†] | ΔAIC[‡] | *ln*L | $\omega_0$ | $\omega_1$ | $\omega_2/\omega_d$ | Null | *P* [df] |
|---|---|---|---|---|---|---|---|
| | | | Parameters | | | | |
| M2a_rel | 17.1 | −30987.99 | 0.01 (60%) | 1 (5%) | 0.19 (35%) | N/A | - |
| CmC_*Characi branch* | 19.05 | −30986.96 | 0.01 (60%) | 1 (5%) | 0.19 (35%) Char Br: 0.20 | M2a_rel | 0.794 [1] |
| CmC_*Characi clade* | 0* | −30977.43 | 0.00 (60%) | 1 (5%) | 0.20 (20%) Char Cl: 0.10 | M2a_rel | **0.000 [1]** |

The foreground partition is listed after the underscore for the clade models and consists of either: the ancestral branch leading to the Characiphysi (*Characi branch; Char Br*) or the entire Characiphysi clade (*Characi clade; Char Cl*). In any partitioning scheme, the entire clade was tested, and all non-foreground data are present in the background partition.

[‡]All ΔAIC values are calculated from the lowest AIC model. The best fit is bolded with an asterisk (*).

[§]$\omega_d$ is the divergent site class, which has a separate value for the foreground and background partitions.

Significant p-values (α ≤0.05) are bolded. Degrees of freedom are given in square brackets after the p-values.

Abbreviations—*ln*L, ln Likelihood; p, p-value; AIC, Akaike information criterion.

DOI: https://doi.org/10.7554/eLife.35957.012

amino acid variation observed in these lineages would be driven by an increase in purifying selection on non-synonymous codons, ultimately reflecting the entrenchment of compensatory amino acid residues by natural selection.

We therefore employed codon-based phylogenetic likelihood methods to test for a relative increase of purifying selection at RH1 sites within 6 Å of site 122, within Tetrapod vs Teleosts, as well as in Characiphysi vs other Teleosts (*Yang, 2007*) (Materials and methods). Using likelihood ratio tests of alternative (Clade model C [(*Bielawski and Yang, 2004*)]) and null (M2a_REL ([*Weadick and Chang, 2012*)]) model analyses of codon substitution rates ($d_N/d_S$) across the RH1 coding-sequence, we identified statistically significant evidence of gene-wide increases in purifying selection within Tetrapods (*Table 3*) and Characiphysi (Table 5) relative to teleosts ((p<0.001)). Sites estimated to be under this increase in purifying selection were those identified in the CmC divergent site class through a Bayes empirical Bayes analysis as previously described (*Castiglione et al., 2017*). Consistent with the fixation of E122 and I122 in tetrapod and Characiphysi RH1, respectively (asterisks, *Figure 2C*), we detected a relative increase of purifying selection on site 122 codons in tetrapod and Characiphysi RH1 relative to that of teleosts (*Figure 2D*; *Tables 3–5*), suggesting that a corresponding increase of purifying selection may have occurred at putatively coevolving sites within the 6 Å radius (*Pollock et al., 2012*; *Goldstein and Pollock, 2017*; *Shah et al., 2015*). No evidence for this was detected at sites 126 and 211, the other members of the TM3-TM5 domain stabilizing the MII active-state (*Table 4*; [(*Choe et al., 2011*)]). Yet within this radius, we found significant evidence for a relative increase of purifying selection in tetrapods and the Characiphysi (relative to teleosts) at several RH1 sites (119, 124, 168; *Figure 2D*; *Table 4*), some of which (sites 119; 168) also displayed significant statistical evidence for covariation (MI) with site 122 in Teleost RH1 (*Figure 2B* vs. 2D; *Table 4*). Furthermore, one of these sites (119) also exceeded the significance threshold in our Bonferroni-corrected phylogenetic tests of correlated evolution with site 122 where p-values were calculated by performing Monte Carlo tests using data from simulations (*Pagel, 1994*) (Materials and methods; *Table 4*). Taken together, these results provide evidence that an increase in purifying selection on non-synonymous codons drove the reduction in amino acid variation at positions coevolving with site 122, and this likely accompanied the fixation of E122 and I122 in tetrapods and the Characiphysi, respectively.

Due to the consistency of these findings with coevolutionary entrenchment, we hypothesized that we could identify fixed residues within this 6 Å radius in Characiphysi RH1 that may be functionally compensatory for the ancient E122I mutation that occurred in the ancestral Characiphysi (*Figure 2A*). Of the RH1 sites displaying significant statistical evidence for covariation (MI) with site 122 in Teleost RH1 (*Figure 2B*; 119, 127, 168), as well as those displaying significant evidence for a relative increase of purifying selection in tetrapods and the Characiphysi (relative to teleosts; 119, 124, 168; *Figure 2D*; *Table 4*) only sites 119, 124 and 127 had fixed amino acid residues in Characiphysi RH1 relative to other Teleosts (asterisks, *Figure 2C*), suggesting this strict conservation pattern may reflect entrenchment due to the fixation of I122. In contrast, despite a statistically significant

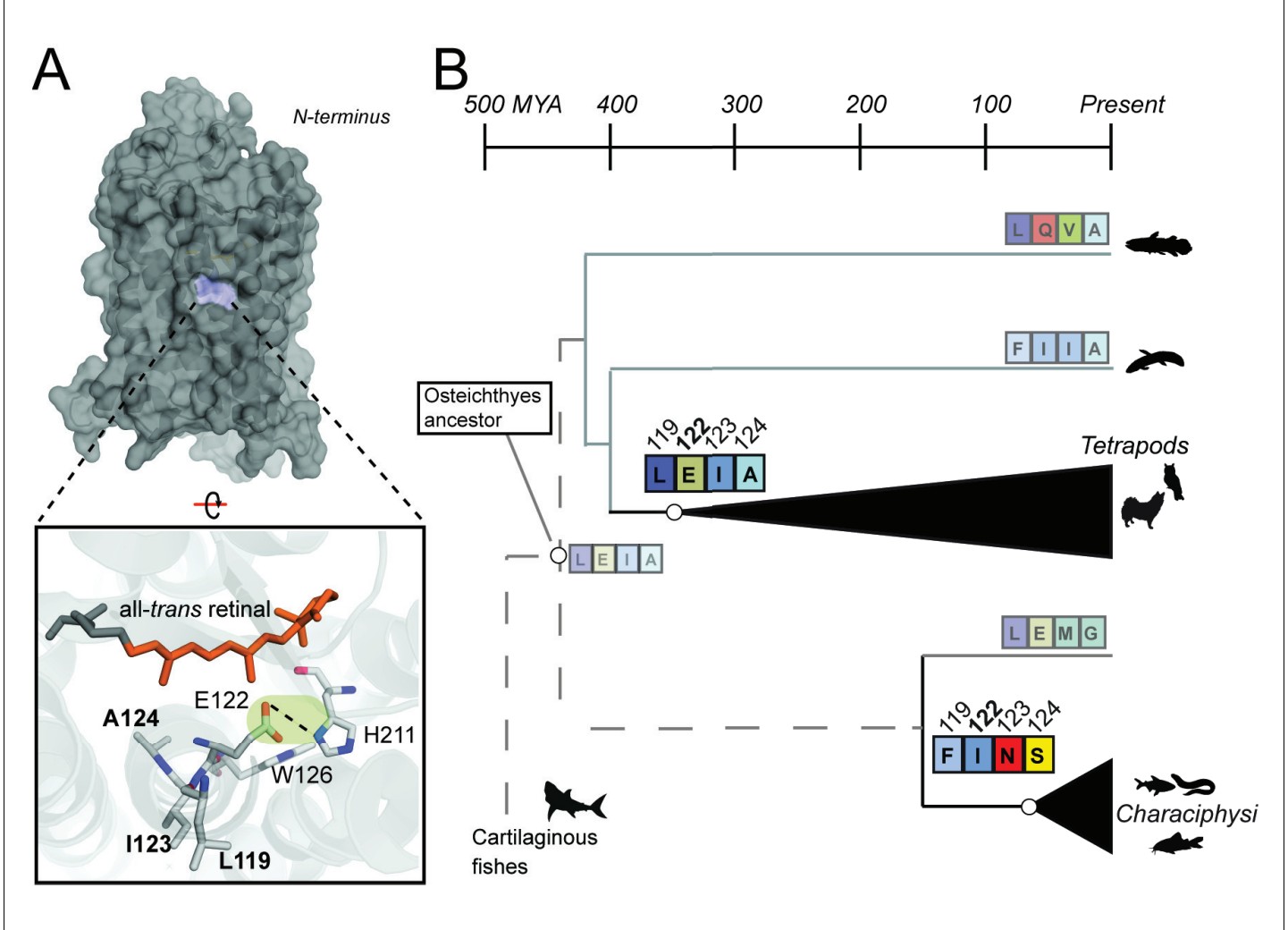

**Figure 3.** Coevolving sites form the LxxEIA and FxxINS motifs. (**A**) Overview of tetrapod RH1 MII rhodopsin crystal structure (*Choe et al., 2011* with coevolving sites. The green highlight and dashed line indicate the stabilizing hydrogen bond between E122-H211. (**B**) Reconstruction of residues at site 122 (*Figure 2—figure supplement 1*) and coevolving positions for ancestral characiphysi, tetrapod and outgroup rhodopsins indicates the entrenchment of two structural motifs centering around site 122 (Materials and methods). The LxxEIA (or LEIA) motif was also predicted as present within the ancestral Osteichthyes. Approximate divergence times are from (*Hedges et al., 2015*.
DOI: https://doi.org/10.7554/eLife.35957.013

increase in purifying selection on non-synonymous codons relative to other Teleosts, site 168 nevertheless displayed amino acid variation in the Characiphysi (T/V168; *Figure 2C, D*), suggesting it may not necessarily play a functionally compensatory role for the ancient E122I mutation, especially since T vs. V168 may be reasonably expected to have biochemically and/or structurally dissimilar effects on this region of the rhodopsin TM3-TM5 microdomain (*Choe et al., 2011*). Conversely, although C127 has been fixed in Characiphysi RH1 relative to other Teleosts (asterisks, *Figure 2C*) and may therefore be functionally important, there was no increase in purifying selection on non-synonymous codons at site 127 relative to Teleosts (*Figure 2D*), suggesting that the fixation of C127 in Characiphysi RH1 may be a historical contingency that does not necessarily reflect intramolecular entrenchment by the ancient E122I mutation (*Goldstein and Pollock, 2017*. Although this same logic ostensibly applies to site 123, unlike C127—a residue shared with some tetrapods (*Figure 2C*; *Supplementary files 1–3*)—we observed a striking fixation of a rare amino acid residue in Characiphysi RH1 (N123, asterisks, *Figure 2C*) which is not, to our knowledge, observed within any vertebrate rhodopsin other than the Characiphysi where it is completely fixed (*Tables 1–2*, *Supplementary files 1–3*), and located between coevolving sites 119, 122 and 124 which are also

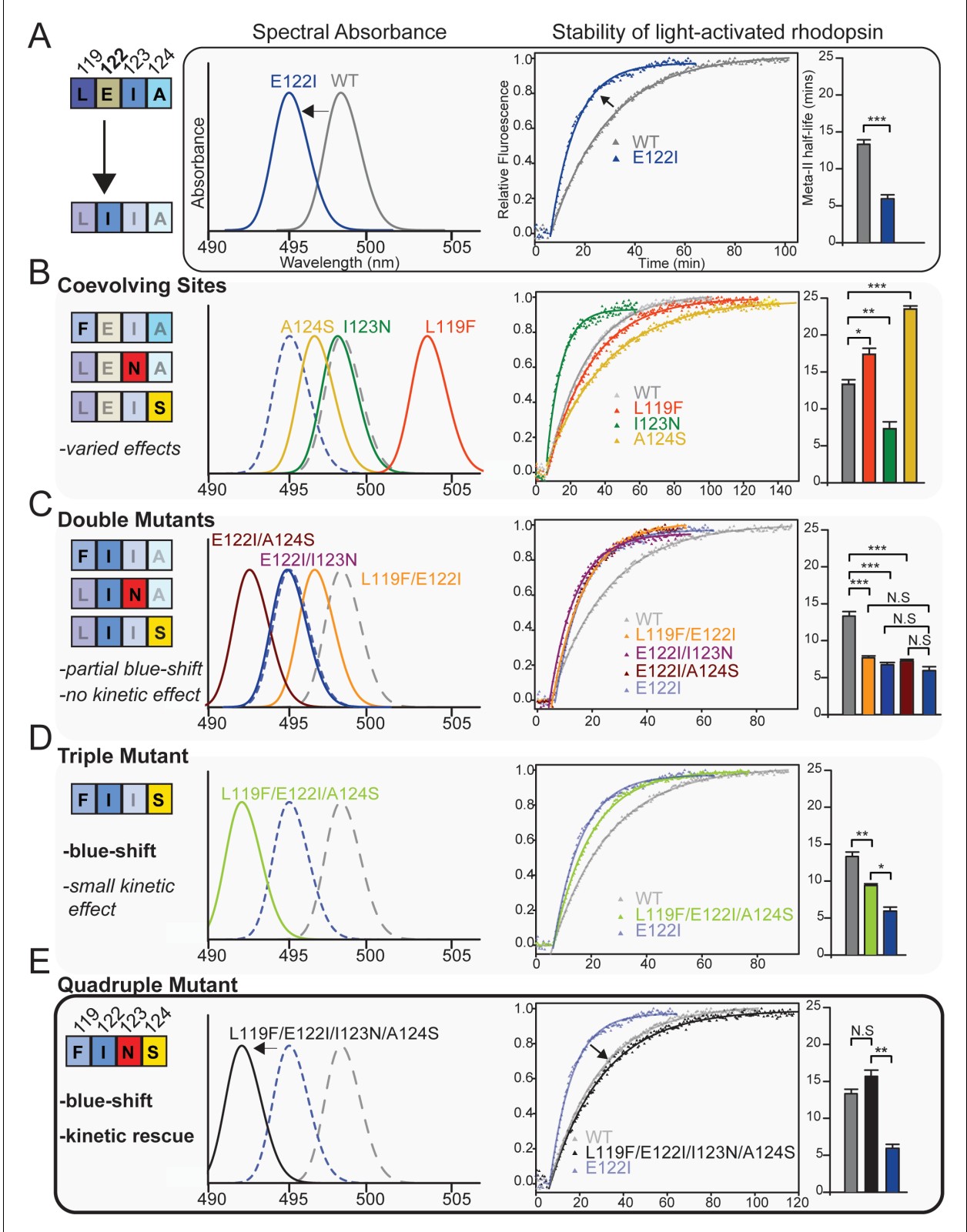

**Figure 4.** Coevolving sites modulate the pleiotropic functional effects of site 122. The LEIA and FINS motifs are convergent solutions for high tetrapod RH1 active state (MII) stability but with different spectral absorbances. (**A**) The introduction of the ancestral cone opsin variant into tetrapod RH1 (E122I) blue-shifts rhodopsin absorbance $\lambda_{MAX}$ and dramatically destabilizes the MII active-conformation (*Figure 4—figure supplement 1*; *Table 6*). Bar graphs show retinal release half-life values. (**B**) Substituting FINS motif residues into coevolving sites have varied effects on rhodopsin spectral tuning

*Figure 4 continued on next page*

*Figure 4 continued*

and the stability of the active-conformation. (C) Within the E122I background, FINS motif substitutions at coevolving sites have marked effects on spectral tuning, but no rescue effect on MII active-conformation stability. (D) Partial incorporation of the FINS motif within tetrapod rhodopsin produces further blue-shifting effects and has a significant but small stabilizing effect within the E122I background. (E) Full incorporation of the FINS motif into tetrapod RH1 maintains the absorbance blue-shift while fully rescuing the destabilizing effects of E122I on tetrapod RH1. Statistically significant differences in MII stability were calculated using two-tailed *t*-tests with unequal variance, with standard error reported in bar graphs (*p<0.05; **p<0.01; ***p<0.001). The number of biological replicates (i.e. separate elutions and/or purifications of rhodopsin) are described in *Table 6*.

DOI: https://doi.org/10.7554/eLife.35957.014

The following figure supplement is available for figure 4:

**Figure supplement 1.** Absorbance spectra of dark-state WT and mutant rhodopsins with wavelength of maximum absorbance ($\lambda_{MAX}$) shown.

DOI: https://doi.org/10.7554/eLife.35957.015

fixed in the Characiphysi (*Figures 2C*; 3A). This unique natural variation is particularly interesting as site 123 has been previously found capable of functionally compensating for human pathogenic mutations (e.g. A164V) disrupting the MII-stabilizing E122-H211 interaction (*Stojanovic et al., 2003*).

Therefore, we decided to focus on sites 119, 123, and 124, two of which (119, 123) are thought to have functional effects via the TM3-TM5 microdomain (*Stojanovic et al., 2003*). Altogether, these sites are located in close proximity to several important structural regions known to affect MII stability, such as N302 of the NPxxY motif, the TM3-TM5 microdomain involving sites 122–211, as well as the all-*trans* retinal binding pocket (*Choe et al., 2011*); *Figure 3A*), suggesting that residues at these positions may form novel structural interactions that could compensate for the destabilizing loss of the E122-H211 hydrogen bond (*Stojanovic et al., 2003*; *Morrow et al., 2017*). Consistent with the entrenchment of compensatory mutations at coevolving sites (*Talavera et al., 2015*; *Pollock et al., 2012*; *Shah et al., 2015*), using ancestral reconstruction we found that sites 119, 122, 123, and 124 are strongly conserved as the LxxEIA (L119-E122-I123-A124; referred to as 'LEIA') and FxxINS motifs (F119-I122-N123-S124; 'FINS') within tetrapod and Characiphysi RH1, respectively, likely since the most recent common ancestor of each lineage, where LEIA is predicted as the ancestral Osteichthyes motif (*Figure 3B*; *Tables 1*, *Supplementary files 1,3*). The maintenance of these two completely different amino acid motifs in Characiphysi and tetrapod RH1 strongly suggests that natural selection has constrained intramolecular interactions at these sites, which we hypothesized to be associated with modulating the pleiotropic functional consequences of sequence variation at site 122.

**Table 6.** Summary of spectroscopic assays on wild-type and mutant rhodopsins.

| | 119-122-123-124 motif | $\lambda_{MAX}$ (nm)[*,†] | Half-life of retinal release[1,2] |
|---|---|---|---|
| Wild-type Bovine rhodopsin | LEIA | 498.2 ± 0.1 (4) | 13.3 ± 0.6 (4) |
| L119F | FEIA | 503.5 ± 0.8 (3) | 17.4 ± 0.8 (3) |
| E122I | LIIA | 495.4 ± 0.2 (4) | 5.93 ± 0.6 (3) |
| I123N | LENA | 498.0 ± 0.3 (3) | 7.31 ± 0.9 (4) |
| A124S | LEIS | 496.6 ± 0.2 (3) | 23.5 ± 0.4 (3) |
| L119F/E122I | FIIA | 496.6 ± 0.4 (3) | 7.68 ± 0.3 (3) |
| E122I/I123N | LINA | 494.8 | 6.47 ± 0.3 (3) |
| E122I/A124S | LIIS | 492.5 ± 0.5 (3) | 7.24 ± 0.2 (3) |
| L119F/E122I/A124S | FIIS | 492.2 ± 0.1 (3) | 9.40 ± 0.2 (3) |
| L119F/E122I/I123N/A124S | FINS | 492.6 ± 0.3 (4) | 15.7 ± 0.8 (3) |

[*]Standard error is shown.

[†]Number of biological replicates (i.e. separate elutions and/or purifications of rhodopsin) is shown in brackets.

DOI: https://doi.org/10.7554/eLife.35957.016

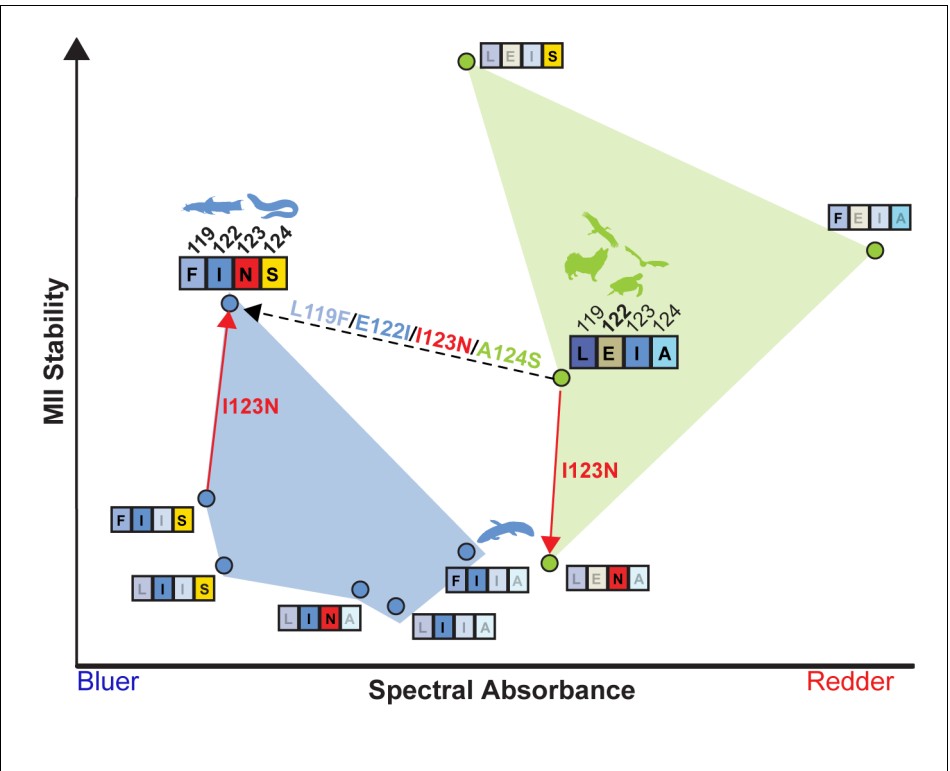

**Figure 5.** LEIA and FINS motifs are alternative solutions for high tetrapod RH1 MII stability within a limited sequence-function landscape. Spectral absorbance ($\lambda_{MAX}$) and stability of the active-conformation (MII) of wild type and mutant tetrapod RH1 with E122 (green) and I122 (blue), respectively. The only natural intermediate between the wild-type tetrapod consensus motif (LEIA) and the wild-type Characiphysi motif (FINS) is 'FIIA' from Lungfish RH1. The mutation I123N has opposite effects on MII stability depending on background sequence (sign-epistasis), which may have closed the LEIA to FINS motif evolutionary trajectory (dashed line) for tetrapod RH1. Although reflecting a limited experimental dataset, these epistatic effects may have created indirect routes to the high MII stability of the FINS motif *via* intermediates with low MII stability.
DOI: https://doi.org/10.7554/eLife.35957.017
The following figure supplement is available for figure 5:

**Figure supplement 1.** Compensatory effects at coevolving sites are mediated by a diversity of possible structural mechanisms.
DOI: https://doi.org/10.7554/eLife.35957.018

## Experimental characterization of natural variation at coevolving sites

We therefore tested the ability of coevolving sites 119, 123 and 124 to affect tetrapod rhodopsin function and the potential for natural variation at these sites to compensate for the destabilizing loss of the E122-H211 hydrogen bond. We conducted site-directed mutagenesis and in vitro expression of mutant rhodopsins using detergent micelles (Materials and methods). This was followed by in vitro functional characterization using spectroscopic absorbance- and fluorescence-based measurements of both $\lambda_{MAX}$ and the stability of the active-state conformation (*Figure 4*; *Figure 4—figure supplement 1*; *Table 6*; Materials and methods), both of which can provide information on relative differences that exist within natural systems (*Schafer et al., 2016*; *Van Eps et al., 2017*; *Schott et al., 2016a*). Tetrapod RH1 with E122I (*Figure 4A*) and other FINS motif single substitutions to the coevolving sites (L119F, I123N, A124S; *Figure 4B*) displayed large shifts in rhodopsin $\lambda_{MAX}$ and MII stability, with two single mutations (L119F, A124S) significantly increasing the stability of the active-conformation but producing opposite spectral tuning effects (*Figure 4B*; *Table 6*). Meanwhile, I123N destabilized the active-conformation almost as dramatically as E122I but produced no spectral tuning effect (*Figure 4B*; *Table 6*). This suggested that FINS substitutions at coevolving sites could functionally compensate for some of the pleiotropic effects of E122I on tetrapod rhodopsin.

We created double and triple mutants representing partial replacements of the LEIA with the FINS motif, which tended to blue-shift $\lambda_{MAX}$ (*Figure 4C–D*; *Table 6*). Yet, none of these intermediates were sufficient to restore WT-levels of MII stability within the COV I122 background (*Figure 4C–D*; *Table 6*). We therefore reasoned that the complete recapitulation of the FINS motif within tetrapod rhodopsin may be required for a full restoration of WT active-conformation stability. We found, incredibly, that the L119F/I123N/A124S triple mutation fully restored the MII stability of E122I tetrapod rhodopsin to WT levels, while even further blue-shifting $\lambda_{MAX}$ relative to E122I (*Figure 4E*; *Table 6*). The LEIA and FINS motifs are therefore two configurations conferring convergent MII stabilities but different spectral sensitivities, with the blue-shifting I122-containing FINS motif likely also decreasing rod dark noise in vivo (*Gozem et al., 2012*; *Yue et al., 2017*).

Our experiments demonstrate that N123, which is not, to our knowledge, observed within any vertebrate rhodopsin other than the Characiphysi (*Tables 12*, *Supplementary files 1–3*) is nevertheless required for a complete rescue of MII stability within the LWS COV I122 background, where it has opposite functional effects depending on E vs. I122 backgrounds (also known as sign-epistasis [*Storz, 2016*; *Weinreich et al., 2006*]) (*Figure 4D–E*; *Figure 5*). Structural analysis of a homology model of the MII active-state structure (Materials and methods; *Figure 5—figure supplement 1*) suggests the conformation of the FINS motif mediates a series of context-dependent structural rearrangements promoting novel interactions (F119 with W161; N123 with N78/T160; *Figure 5—figure supplement 1*) that can interact with existing GPCR hydrogen bond networks known to stabilize the MII active conformation (S124 with D83-S298- N302; *Figure 5—figure supplement 1*; [*Choe et al., 2011*]). These epistatic structural interactions produce correspondingly variable pleiotropic effects on RH1 spectral absorbance and MII stability (*Figure 4*), which were consistent with patterns of natural sequence variation at these positions across vertebrate rhodopsins. Using these patterns of naturally occurring sequence variation, we could successfully navigate a complex sequence-function landscape (*Figure 5*) to engineer the spectral *and* non-spectral functions of rhodopsin simultaneously.

## Discussion

We questioned why E122, a residue that diminishes rod photosensitivity, was retained in the evolution of all tetrapod rod pigments. We investigated if it was possible to engineer a tetrapod RH1 with high active-state conformational stability *without* E122. We uncovered a natural solution—over 150 million years ago the 'FINS' motif originated within the rhodopsin of an ancestral population of the Characiphysi freshwater fish lineage. Although we explore here only a limited subsection of the total sequence-function space of tetrapod vs. Characiphysi RH1 (notably excluding sites 127 and 168 in our experimental analyses), this natural variation, nevertheless, inspired us to engineer a synthetic alternative that nature never produced: a tetrapod RH1 with high MII stability *without* E122, resulting in a blue-shifted pigment predicted to increase rod photosensitivity in vivo. These results, along with recent advances in molecular evolutionary theory, and studies of rhodopsin biochemistry, suggest a plausible model of why the FINS motif might have been the road less traveled in evolutionary history.

### Physiological relevance of MII stability—a proposed role in photoprotection in the eye

Did a physiological advantage related to high MII stability drive the fixation of the LEIA and FINS motifs? Consistent with predictions of intramolecular coevolutionary theory (*Talavera et al., 2015*; *Pollock et al., 2012*; *Shah et al., 2015*), evolutionary trajectories from the LEIA to FINS motifs must pass through sub-optimal sequence-function intermediates which include variants associated with active-state instability and human rhodopsin disease phenotypes (e.g. A164V) (*Stojanovic et al., 2003*) (*Figure 5*). Similar to E122I, disease variants such as A164V are likely pathogenic through disruption of the E122-H211 hydrogen bond, which has been shown to stabilize the active-state conformation but can be affected indirectly through mutations at nearby sites 119 and 123 (*Imai et al., 1997*; *Stojanovic et al., 2003*; *Morrow and Chang, 2015*. Although correlations between dark-state stability and active-state (MII) stability have been recently postulated (*Kojima et al., 2017*), there exists substantial conformational differences in the TM3-TM5 region thought to stabilize both structures, including the reconfiguration of E122-H211 and E122-W126 hydrogen bonds upon light

activation (*Choe et al., 2011*; *Ahuja et al., 2009*; *Okada et al., 2004*; *Lin and Sakmar, 1996*). While it is unclear if such structural differences exist within the cone opsins, the lack of E122 (e.g. Q122 in Rh2 (except the lamprey [(*Lin et al., 2017*; *Davies et al., 2007*)), I122 in LWS (*Lamb et al., 2007*; *Carleton et al., 2005*)) strongly suggests that natural selection has prioritized dark-state stability over MII stability within the cone opsins, which may be related to mitigating the high noise of cone photoreceptors, especially in red-shifted LWS (*Gozem et al., 2012*; *Kefalov et al., 2003*; *Imai et al., 1997*; *Chen et al., 2012a*; *Kefalov et al., 2005*). By contrast, in tetrapod rhodopsins, E122 predominates, increasing MII stability while red shifting spectral absorbance, and therefore also decreasing in vivo rod photosensitivity (*Gozem et al., 2012*; *Yue et al., 2017*). This suggests that selection has maintained E122, and therefore the stability of the MII active-conformation, for reasons distinct from those maintaining the stability of the dark-state conformation, which modulates rod photosensitivity.

Why has the increased rod photosensitivity conferred by COV at site 122 been sacrificed in all tetrapods? In addition to setting the limit on rod photosensitivity (*Baylor et al., 1980*), rhodopsin is also associated with light-induced photodamage (*Grimm et al., 2000*; *Williams and Howell, 1983*), where retinal susceptibility strongly correlates with rhodopsin expression levels, and can be altered through ambient lighting conditions in some animals (*Rózanowska and Sarna, 2005*; *Organisciak and Vaughan, 2010*). Below we describe the mounting indirect evidence that the high stability and long decay of the rhodopsin MII active conformation may be a photoprotective mechanism against light-induced retinal damage (*Sommer et al., 2014*; *Imai et al., 2005*), and we postulate that this likely accompanied the evolution of dim-light vision.

First, one of the most promising strategies to increase retinal resistance to photodamage is to slow the rate of rhodopsin regeneration, which can be achieved *via* mutations or molecules inhibiting the normal functioning of visual cycle proteins responsible for synthesizing 11-*cis* retinal (11CR) (*Wenzel et al., 2001*; *Saari et al., 2001*; *Mandal et al., 2011*). This can also be done through blocking rhodopsin regeneration and binding of 11CR (*Radu et al., 2003*; *Sieving et al., 2001*), reducing the light-dependent accumulation of atRAL condensation products such as diretinoid-pyridinium-ethanolamine (A2E), which contributes to lipofuscin deposits in the retinal pigment epithelium associated with human retinal diseases (*Maeda et al., 2009*; *Chen et al., 2012b*; *Radu et al., 2003*; *Sparrow, 2003*). Importantly, whether rhodopsin will bind available 11CR, or atRAL is dictated by the conformational selectivity of the rhodopsin dark- and active-state (MII) conformations, respectively (*Schafer et al., 2016*; *Schafer and Farrens, 2015*; *Chen et al., 2012a*)—a new finding consistent with previous observations that rhodopsins with high MII stability tend to also have slowed 11CR regeneration rates, which is a key distinguishing feature from the cone opsins (*Chen et al., 2012a*; *Imai et al., 2005*). These observations suggest that rhodopsin conformational selectivity may be an overlooked functional specialization of dim-light vision—one that may be associated with rates of regeneration and therefore photoprotection in the eye.

Although multiple molecular mechanisms within the visual cycle appear to have evolved to prevent the accumulation of toxic atRAL (*Rózanowska and Sarna, 2005*; *Chen et al., 2012c*) as well as excess 11CR (*Quazi and Molday, 2014*, a possible role for rhodopsin's intrinsic conformational selectivity in sequestering these retinal ligands has been mostly overlooked. Recent advances in rhodopsin biochemistry suggest that such a photoprotective mechanism may indeed exist. In contrast to cones, which have an expanded retinoid recycling capacity (*Wang and Kefalov, 2011*; *Tsybovsky and Palczewski, 2015*), in rods atRAL clearance is limited by the activity of retinal dehydrogenases (RDH) (*Saari et al., 1998*; *Chen et al., 2012b*; *Chen et al., 2012c*), and can transiently accumulate to toxic levels (*Sommer et al., 2014* causing light induced-retinopathy through a variety of mechanisms involving oxidative stress (*Maeda et al., 2009*; *Chen et al., 2012b*. Unlike cone opsins, which have low active-conformation stability, the rhodopsin active-conformation is highly stable due in large part to the evolution of E122 (*Imai et al., 1997*), and when phosphorylated and bound to rod arrestin, contains a binding affinity for atRAL sufficient for sequestration and reduction below toxic levels (*Sommer et al., 2014*; *Rózanowska and Sarna, 2005*). Indeed, it is now known that elevated atRAL concentrations will increase atRAL re-uptake by active-conformation rhodopsin in vitro (*Schafer et al., 2016*), and in bright light levels when the risk of photodamage and atRAL levels are highest (*Rózanowska and Sarna, 2005*; *Organisciak and Vaughan, 2010*, this process is further promoted by the constitutive binding of arrestin to the active rhodopsin conformation, which importantly does not block RDH access to atRAL (*Gurevich et al., 2011*; *Sommer et al., 2012*). This

proposed survival mechanism not only precludes $G_t$ signaling but also promotes re-uptake of atRAL by rhodopsin, therefore delaying MII decay and regeneration of the dark state, and enhancing atRAL sequestration (*Sommer et al., 2012*; *Sommer and Farrens, 2006*). Although the other rhodopsin in the homodimer bound by arrestin is likely free to decay to the inactive conformation, permitting regeneration with 11CR (*Sommer et al., 2012*; *Schafer and Farrens, 2015*; *Beyrière et al., 2015*), a higher intrinsic stability of the active conformation would not only likely delay regeneration with 11CR, but also likely push the equilibrium toward atRAL re-uptake —a process that could be even further promoted if atRAL levels are high (*Sommer et al., 2012*; *Schafer et al., 2016*), such as within dark-adapted animals exposed to bright flashes (*Saari et al., 1998*; *Lee et al., 2010*, and within disease models where atRAL clearance is delayed (*Maeda et al., 2009*; *Chen et al., 2012c*. The evolution of high conformational selectivity of active-state rhodopsin for atRAL—a distinguishing feature from the cone opsinsmay therefore play a key role within these putatively photoprotective ternary complexes, which have been previously proposed to provide an atRAL sink for rods in bright light (*Sommer et al., 2014*).

While only detailed experimental investigations can determine the relationships between rhodopsin regeneration rates, atRAL-associated photodamage, and the recently expanded ensemble of spectrally identical MII conformational substates (*Van Eps et al., 2017*, a putative photoprotective role for the intrinsic stability of the rhodopsin MII active conformation would imply the presence of strong rhodopsin functional constraints *in addition* to those canonical constraints associated with rod photosensitivity. This model is consistent with the fact that high rod photosensitivity and susceptibility to photodamage appear to be a trade-off that accompanied the evolution of rhodopsin-mediated dim-light vision (*Grimm et al., 2000*; *Williams and Howell, 1983*90,91. Indeed, a trade-off model of rhodopsin evolution may clarify why the experimental relevance of long MII decay still remains unclear, as the focus has been mostly on mutational effects to photosensitivity, rather than photodamage ( *Kojima et al., 2014*; *Imai et al., 2007*). Below, we outline a trade-off model of rhodopsin evolution, and describe in detail how it may help to unravel the paradoxical distribution of natural sequence variation at rhodopsin site 122.

## Trade-offs between rhodopsin-mediated photosensitivity and photoprotection may explain the E122 paradox

As discussed above, tetrapod susceptibility to photodamage is a necessary side-effect of rhodopsin-mediated dim-light vision (*Grimm et al., 2000*; ), yet it has been an often-overlooked possibility that the functional constraints governing rhodopsin evolution could have also been shaped by those associated with rhodopsin-mediated photodamage, which induces oxidative stress leading to retinal degenerative diseases via toxic atRAL (*Tsybovsky and Palczewski, 2015*; *Sommer et al., 2014*; *Różanowska and Sarna, 2005*; *Wenzel et al., 2005*). By delaying both atRAL release and 11CR binding, high rhodopsin MII stability could provide an additional protective mechanism against rhodopsin-mediated photodamage outside the visual cycle—one which could be modulated parsimoniously in response to light conditions through mutations altering MII stability (*Dungan and Chang, 2017*; *Gutierrez et al., 2018*; *Hauser et al., 2017b*. Our model would therefore predict that the evolution of rhodopsin after divergence from the cone opsins involved unique functional specializations for both photosensitivity *and* photoprotection. These photodamage-related constraints associated with the evolution of dim-light vision may have been especially relevant within dim-light adapted animals with high levels of rhodopsin and an increased susceptibility to photodamage (*Różanowska and Sarna, 2005*; *Organisciak and Vaughan, 2010* where exposures to bright light flashes can dramatically increase toxic ATR accumulation levels, leading to photoreceptor degeneration (*Saari et al., 1998*; *Chen et al., 2012b*).

Interestingly, due to environmental differences, variation in the selective constraints associated with rhodopsin's role in photosensitivity vs. photodamage may explain the paradoxical ecological patterns associated with natural variation at site 122. Specifically, our results demonstrate that the only visual ecologies where the selective constraints on E122 are repeatedly relaxed across the phylogeny is within the constant darkness of deep-dwelling fish environments (*Hunt et al., 2001*; *Yokoyama et al., 1999*)—the natural system where one might expect the fitness effects of the rhodopsin-mediated trade-off between photosensitivity and photoprotection to drastically shift, as there is likely little photodamage risk for fishes living below 1000 m in near-permanent darkness (*Denton, 1990*). All whales by contrast—some of which routinely dive into complete darkness at

depths near 2000 m (e.g. the sperm whale) (*Denton, 1990*; *Watkins et al., 1993*)—strictly maintain E122, thereby forgoing the photosensitivity increase conferred by COV at site 122 that would otherwise likely prove advantageous within these dark marine environments (*Hunt et al., 2001*; *Yue et al., 2017*; *Yokoyama et al., 1999*). Yet, unlike deep dwelling fishes, all whales must resurface, suggesting that photoprotection-associated constraints may be maintaining E122 despite the cost of decreased photosensitivity: a prediction consistent with our model and with increases to MII stability as a key feature of whale evolution (*Dungan and Chang, 2017*. In contrast, in Characiphysi fishes, the FINS motif evolved—a novel molecular mechanism likely increasing rod photosensitivity *without* the consequent trade-off on MII stability. Although this may be related to mitigating the increased dark noise that can arise as consequence of spectral tuning to some red-shifted freshwater environments (*Gozem et al., 2012*; *Van Nynatten et al., 2015*), this remains unclear, as the ancestral condition is uncertain, and the distribution of characiphysian fishes occur in a wide range of environments (*Castiglione et al., 2017*; *Chen et al., 2013*). Whether other sequence motifs within this network represent different 'tuning solutions' for the visual system across different environments is unknown, yet the possible permutations appear to have been dramatically limited by a combination of natural selection, historical contingency and epistasis (*Tables 1–2*) (*Storz, 2016*). This would be an interesting avenue of future investigation.

In tetrapods, none of these coevolutionary motifs include COV at site 122 (*Tables 1*, *Supplementary files 1–2*). This suggests that tetrapods have been confined to a local optimum (E122), which makes it tempting to speculate that this evolutionary constraint could only be maintained by the existence of a strongly detrimental pleiotropic effect, which we propose to be that of rhodopsin-mediated photodamage. Potential caveats to this theory include the existence of subterranean tetrapods maintaining E122 (*Table 1*)—a system where one might expect the putative photodamage-associated constraints on rhodopsin to relax, as may have occurred within a variety of deep-sea fishes (although it remains unclear if increases to photosensitivity would even be prioritized within these animals if the putative constraints on photoprotection were indeed relaxed (*Partha et al., 2017*)). Although speculative, our trade-off model of rhodopsin evolution, combined with fitness landscape theory (*Hartl, 2014*) could potentially explain why the evolutionary trajectories between the LEIA and FINS motifs have been traversed by some freshwater fishes, but never by a tetrapod lineage.

## Exploring inferred fitness landscapes using natural variation

Evolutionary pathways often include compensatory mutations (*Talavera et al., 2015*; *Pollock et al., 2012*; *Shah et al., 2015*; *Tokuriki et al., 2008*), where adaptive mutations are permitted by non-adaptive neutral mutations (*Pál and Papp, 2017*; *Starr et al., 2017*; *Tarvin et al., 2017*) (also known as 'pre-adaptations' [*Pál and Papp, 2017*], or 'pre-adjustments' [(*Goldstein and Pollock, 2017*)) and this contingency opens new evolutionary paths by accommodating the subsequent mutational perturbations to protein activity and/or stability (*Tokuriki and Tawfik, 2009*; *Ivankov et al., 2014*; *DePristo et al., 2005*). Similarly, we find that a triple mutation (L119F/I123N/A124S) would be required to functionally compensate for the detrimental effects of E122I on tetrapod rhodopsin active-conformation stability. Yet, one of these 'pre-adaptations' (I123N) is as destabilizing as E122I itself, and displays strong sign-epistasis (*Figure 4*), which may be sufficient to close the evolutionary trajectory leading from the LEIA to FINS motifs (*Storz, 2016*; *Weinreich et al., 2005*; *Poelwijk et al., 2007*). Accordingly, a wide variety of indirect evolutionary paths may cut through these valleys to access fitness peaks (*Wu et al., 2016*; *Starr et al., 2017*; *Weinreich et al., 2006*)—a scenario which has not been extensively characterized in protein systems evolving within natural environments (*Pál and Papp, 2017*; *Hartl, 2014*).

Detrimental intermediates and historical contingency (*Pál and Papp, 2017*; *Wu et al., 2016*; *Starr et al., 2017*; *Weinreich et al., 2006*; *Palmer et al., 2015*) may ultimately explain why the road to the FINS motif was less travelled by in evolutionary history; although tetrapods could have in theory evolved both higher rod photosensitivity *and* high MII stability via the FINS motif (*Gozem et al., 2012*; *Yue et al., 2017*; *Imai et al., 1997*), the sign-epistasis of site 123 (*Figure 4*) may have constrained tetrapod RH1 to the local fitness optima of E122 and the LEIA motif (*Storz, 2016*; *Weinreich et al., 2005*; *Poelwijk et al., 2007*). E122 as a historical contingency may have promoted MII-mediated photoprotection within terrestrial environments and was therefore entrenched by purifying selection pressures and epistatic interactions with nearby sites (*Pollock et al., 2012*;

*Goldstein and Pollock, 2017*; *Shah et al., 2015*). E122 may therefore be a 'molecular springboard' (*Pál and Papp, 2017* for reaching higher levels of MII stability and photoprotection that COV could not potentiate (*Dungan and Chang, 2017*; *Hauser et al., 2017b*; *Gutierrez et al., 2018*) indeed, E122/S124 is nearly six-fold more stable than I122/S124 (*Figure 4*). By contrast, temporal and spatial variation in fish visual ecologies (*Hauser and Chang, 2017a*; *Bowmaker, 2008* may have opened up the indirect trajectories containing low MII stability (*Pál and Papp, 2017*; *Ogbunugafor et al., 2016*; *Steinberg and Ostermeier, 2016* created by the epistasis of the coevolutionary network (*Wu et al., 2016*; *Palmer et al., 2015*), as evidenced by the fact that selection has allowed multiple fish lineages to innovate at this coevolving network through amino acid variation that may promote photosensitivity instead of MII stability (*Hunt et al., 2001*; *Yue et al., 2017*; *Yokoyama et al., 1999*); *Table 2*, *Supplementary file 2*). Although speculative, this implies that changing environmental constraints within the ancestral Characiphysi population (*Chen et al., 2013*) may have bridged the evolutionary valleys between the LEIA and FINS motifs, as has been observed in some experimental studies on bacterial enzymes mediating antibiotic resistance (*Ogbunugafor et al., 2016*; *Steinberg and Oster-meier, 2016*). The specific environmental differences that may be responsible for opening and closing these alternative evolutionary trajectories within fishes remain to be identified and would be an interesting subject of future investigation. It is important to note that although the sequence-function landscape of site 122 is likely more complex than what we demonstrated here, recent studies from ours and other groups have begun unravelling important epistatic interactions among residues at four-site motifs (*Wu et al., 2016*; *Starr et al., 2017*; *Tarvin et al., 2017*).

We therefore present a powerful integrative approach for the exploration of inferred fitness landscapes using natural variation. This has generated multiple insights. First, our results strongly suggest that E122 was not necessary for the evolution of high MII stability, and therefore expands on previous work demonstrating site 122 as an evolutionary dynamic determinant of visual pigment spectral and non-spectral properties (*Hunt et al., 2001*; *Yue et al., 2017*; *Imai et al., 1997*; *Yokoyama et al., 1999*; *Imai et al., 2007*). Second, this further argues that novel sequence-function solutions in proteins (*McMurrough et al., 2014*; *Mateu and Fersht, 1999*; *Tarvin et al., 2017*) can be discovered by integrating genetic and ecological information to reveal ancient evolutionary trajectories (*Liebeskind et al., 2015*; *McTavish et al., 2013*). These evolutionary solutions may be otherwise unpredictable from biophysical perspectives (*Starr and Thornton, 2017*; *Sailer and Harms, 2017*; *Otwinowski and Plotkin, 2014*) where the accuracy of computational models remains limited to describing changes in protein stability (*Goldenzweig and Fleishman, 2018*; *Echave et al., 2016*; *Goldstein and Pollock, 2017*), rather than the adaptive shifts in protein function and trade-offs *with* stability—a scenario likely widespread in natural systems (*Pál and Papp, 2017*; *McMurrough et al., 2014*; *Mateu and Fersht, 1999*; *Tarvin et al., 2017*; *DePristo et al., 2005*). Finally, our work suggests that even within biological systems as complex as that of animal vision, the existence of novel biophysical *and* ecological constraints can still be elucidated through comparative analyses of natural variation.

## Materials and methods

### Key resources table

| Reagent type (species) or resource | Designation | Source or reference | Identifiers | Additional information |
|---|---|---|---|---|
| gene (*Bos taurus*) | RH1(Rho) | N/A | *Accession: M12689* | |
| cell line (*Homo sapiens*) | HEK293T | Dr. David Hampson, University of Toronto | | Authenticated by STR profiling |
| transfected construct | pIRES-hrGFP II | Stratagene | | |
| antibody | 1D4 monoclonal antibody | doi: 10.1007/978-1-4939-1034-2_1 | | fixed to Ultralink Resin (5mg 1D4:7mL Resin) |
| commercial assay or kit | Lipofectamine 2000 | ThermoFisher Scientific | *Catalog Number: 11668019* | |

*Continued on next page*

*Continued*

| Reagent type (species) or resource | Designation | Source or reference | Identifiers | Additional information |
|---|---|---|---|---|
| Commercial assay or kit | Ultralink Hydrazide Resin | ThermoFisher Scientific | *Catalog Number: 53149* | |
| Chemical compound, drug | 11-*cis* retinal | other | | Dr. Rosalie Crouch, Medical University of South Carolina |
| sSoftware, algorithm | PAML 4.7 | https://doi.org/10.1093 /molbev/msm088 | | |
| sSoftware, algorithm | MISTIC | doi: 10.1093/nar/gkt427 | | |
| Software, algorithm | MODELLER | doi: 10.1002/cpbi.3 | | |

## Dataset assembly

Rhodopsin-coding sequences (*rh1*) originating from Teleost fishes, Tetrapods, and other vertebrate outgroups (*Supplementary file 1–3*) were obtained from GenBank using BlastPhyMe (*Schott et al., 2016b*). Teleost fish *rh1* sequences were sampled from all available phylogenetic orders denoted in *Betancur et al., 2013*. Tetrapod *rh1* sequences were sampled from all major phylogenetic groupings (*Figure 1—figure supplement 1*) (*Hedges et al., 2015*; *Foley et al., 2016*; *Prum et al., 2015*; *Amemiya et al., 2013*, as described previously (*Hauser et al., 2016*). Rh1 alignments were generated using PRANK codon alignment (*Löytynoja and Goldman, 2008*. The final *rh1* alignment encoded for rhodopsin amino acid residues 42 – 307 (bovine RH1 numbering), inclusively, where for mutual information analyses gaps were trimmed from the beginning and end of the alignment, resulting in a shorter alignment (residues 53– 302). In both instances, the alignments used for bioinformatic analysis encompassed the entire seven-transmembrane domain of rhodopsin. Using this alignment, we constructed three separate *rh1* datasets for phylogenetic analysis: (1) Tetrapods (n = 86; *Supplementary file 1*); (2) Teleost fishes (n = 119; *Supplementary file 2*); (3) Vertebrate (n = 209) which included (1) and (2) in addition to outgroups. For each dataset, a species tree was constructed by reference to established relationships for Tetrapods (*Hedges et al., 2015*; *Foley et al., 2016*; *Prum et al., 2015*; *Amemiya et al., 2013*) and Teleosts (*Betancur et al., 2013*). The Vertebrate phylogeny was assembled by adding non-tetrapod Sarcopterygian outgroups to the Tetrapod phylogeny, combining this with the Teleost phylogeny, and then adding cartilaginous fish

**Table 7.** Analyses of selection on Vertebrate rhodopsin (*rh1*) using PAML random sites models.

| Model | *ln*L | Parameters[1] | | | Null | P [df][2] | Δ AIC[§] |
|---|---|---|---|---|---|---|---|
| | | $\omega_0$/p | $\omega_1$/q | $\omega_2$/$\omega_p$ | | | |
| M0 | −49624.89 | 0.08 | - | - | N/A | - | 5516.80 |
| M1a | −48355.44 | 0.05 (89%) | 1.00 (11%) | - | M0 | 0.000 [1] | 2979.91 |
| M2a | −48355.44 | 0.05 (89%) | 1.00 (3%) | 1.00 (8%) | M1a | 1 [2] | 2983.91 |
| M3 | −47104.84 | 0.01 (58%) | 0.11 (30%) | 0.44 (12%) | M0 | **0.000 [4]** | 484.71 |
| M7 | −46906.24 | 0.24 | 1.19 | - | N/A | - | 81.51 |
| M8a | −46864.60 | 0.32 | 3.10 | 1.00 | N/A | - | 0.230 |
| M8 | −46863.49 | 0.32 | 2.94 | 1.14 | M7 | **0.000 [2]** | 0* |
| | | | | | M8a | 0.135 [1] | |

*1$\omega$ values of each site class are shown are shown for model M0-M3 ($\omega_0$– $\omega_2$) with the proportion of each site class in parentheses. For M7 and M8, the shape parameters, p and q, which describe the beta distribution are listed instead. In addition, the $\omega$ value for the positively selected site class ($\omega_p$, with the proportion of sites in parentheses) is shown for M8.

2Significant p-values (α ≤0.05) are bolded. Degrees of freedom are given in square brackets after the p-values.

3#Model fits were assessed by Akaike information criterion differences to the best fitting model (asterisk).

Abbreviations—*ln*L, ln Likelihood; p, p-value; N/A, not applicable.

DOI: https://doi.org/10.7554/eLife.35957.019

**Table 8.** Analyses of selection on Teleost rhodopsin (*rh1*) using PAML random sites models.

| Model | *In*L | Parameters1[†] $\omega_0$/p | $\omega_1$/q | $\omega_2/\omega_p$ | Null | P [df][†] | Δ AIC[§] |
|---|---|---|---|---|---|---|---|
| M0 | −32949.46 | 0.10 | - | - | N/A | - | 4489.79 |
| M1a | −31605.10 | 0.05 (86%) | 1.00 (14%) | - | M0 | **0.000 [1]** | 1803.10 |
| M2a | −31605.10 | 0.05 (86%) | 1.00 (10%) | 1.00 (4%) | M1a | 1 [2] | 1807.10 |
| M3 | −30887.40 | 0.01 (58%) | 0.13 (29%) | 0.57 (13%) | M0 | **0.000 [4]** | 373.67 |
| M8a | −30790.44 | 0.28 | 3.05 | 1.00 | N/A | - | 173.76 |
| M7 | −30767.11 | 0.19 | 0.68 | - | N/A | - | 127.10 |
| M8 | −30702.57 | 0.25 | 1.62 | 1.92 | M7 | **0.000 [2]** | 0* |
| | | | | | M8a | **0.000 [1]** | |

1$\omega$ values of each site class are shown are shown for model M0-M3 ($\omega_0$– $\omega_2$) with the proportion of each site class in parentheses. For M7 and M8, the shape parameters, p and q, which describe the beta distribution are listed instead. In addition, the $\omega$ value for the positively selected site class ($\omega_p$, with the proportion of sites in parentheses) is shown for M8.

2Significant p-values (α ≤0.05) are bolded. Degrees of freedom are given in square brackets after the p-values.

3#§Model fits were assessed by Akaike information criterion differences to the best fitting model (asterisk).

Abbreviations—*In*L, ln Likelihood; p, p-value; N/A, not applicable.

DOI: https://doi.org/10.7554/eLife.35957.020

outgroups, all according to species relationships (*Figure 1—figure supplement 2*) (*Betancur et al., 2013*; *Amemiya et al., 2013*122,125. These phylogenies were used in subsequent computational analyses. We also constructed a Characiphysi *rh1* dataset, representing wide phylogenetic sampling (*Supplementary File 3*), with an *rh1* alignment encoding for residues 42– 307 as that described above, and where a species tree (*Figure 2—figure supplement 1*) was constructed by reference to established relationships (*Chen et al., 2013* and references therein).

## Analyses of intramolecular coevolution

We took a multifaceted approach toward detecting sites coevolving with site 122, corroborating our phylogenetic tests of evolutionary rates (dN/dS) (*Yang, 2007*) with phylogenetically corrected statistical tests of amino acid covariation (*Simonetti et al., 2013*), and phylogenetic analyses of correlated evolutionary patterns in amino acid substitutions (*Pagel, 1994*. We used these three approaches to search for evidence of coevolution between rhodopsin site 122 and sites within a 6 Å radius within the MII active-conformation crystal structure (*Choe et al., 2011*).

We used codon models of molecular evolution from the PAML 4.7 software package (*Yang, 2007*) to identify evidence of increased purifying selection in rhodopsin-coding sequences (*rh1*). First, we estimated the evolutionary rates ($d_N/d_S$) within each *rh1* dataset (Teleosts, Tetrapods, Vertebrates, Characiphysi) using the random sites models (M1, M2, M3, M7, M8) implemented in the CODEML program. This required pruning the outgroups from the Teleost and Tetrapod datasets. Site-specific evolutionary rates were obtained from M8, which was the best fitting model in each dataset as assessed by differences in Akaike information criterion (*Tables 7–10*). Next, we employed PAML Clade models (*Bielawski and Yang, 2004* to explicitly test for long-term shifts in evolutionary rates ($d_N/d_S$) between foreground and background branches or clades within the rhodopsin datasets. In any partitioning scheme, all non-foreground data are present in the background partition. The foreground partition is listed after the underscore for the clade models (e.g. CmC_*foreground*). CmC analyses tested for long-term shifts in purifying selection between: tetrapod and teleost clades within the Vertebrate dataset (*Table 3*); the branch leading to the tetrapod clade within the Vertebrate dataset (*Table 3*); and the Characiphysi clade and the branch leading to the clade within the Teleost dataset (*Table 5*). M2aREL was used as the null model (*Weadick and Chang, 2012*). For all PAML models, multiple runs with different starting priors were carried out to check for the convergence of parameter estimates. Significant differences in model fits we determined by likelihood ratio-tests.

Statistical tests of covariation (e.g. Mutual Information; MI) are an approximate measure for identifying coevolving sites in alignments of homologous protein families, but can have high false-

**Table 9.** Analyses of selection on Tetrapod rhodopsin (*rh1*) using PAML random sites models.

| Model | *In*L | Parameters[†] | | | Null | P [df][‡] | Δ AIC[§] |
|---|---|---|---|---|---|---|---|
| | | $\omega_0$/p | $\omega_1$/q | $\omega_2$/$\omega_p$ | | | |
| M0 | −15541.64 | 0.05 | - | - | N/A | - | 1154.78 |
| M1a | −15345.33 | 0.03 (93%) | 1.00 (7%) | - | M0 | [1] | 764.17 |
| M2a | −15345.33 | 0.03 (93%) | 1.00 (0%) | 1.00 (7%) | M1a | 1 [2] | 768.17 |
| M3 | −14981.40 | 0.00 (61%) | 0.06 (28%) | 0.29 (11%) | M0 | **0.000** [4] | 42.31 |
| M7 | −14971.78 | 0.19 | 2.76 | - | N/A | - | 17.10 |
| M8 | −14961.25 | 0.20 | 3.55 | 1.00 | M7 | **0.000** [2] | 0* |

1$\omega$ values of each site class are shown are shown for model M0-M3 ($\omega_0$– $\omega_2$) with the proportion of each site class in parentheses. For M7 and M8, the shape parameters, p and q, which describe the beta distribution are listed instead. In addition, the $\omega$ value for the positively selected site class ($\omega_p$, with the proportion of sites in parentheses) is shown for M8.

2Significant p-values (α ≤0.05) are bolded. Degrees of freedom are given in square brackets after the p-values.

3#Model fits were assessed by Akaike information criterion differences to the best fitting model (bolded asterisk).

Abbreviations—*In*L, ln Likelihood; p, p-value; N/A, not applicable.

DOI: https://doi.org/10.7554/eLife.35957.021

positive rates due to sampling bias and random background effects (*Ashenberg and Laub, 2013*; *Talavera et al., 2015*), especially if there is a lack of phylogenetic correction (*Simonetti et al., 2013*; *Dunn et al., 2008*). Nevertheless, MI methods appear able to detect sites of functional importance that are close in proximity to each other (*Ashenberg and Laub, 2013*; *Talavera et al., 2015*). Given all these factors, we decided to employ MI analyses within our dataset only as a qualitative guide to provide additional insight into the putative coevolutionary dynamics within Vertebrate RH1, and to potentially corroborate our molecular evolution analyses since overlap between evolutionary rates and statistical covariation of amino acids has been described in detail (*Talavera et al., 2015*. Since MI is usually employed within large protein family datasets, rather than intrafamily comparisons (*Ashenberg and Laub, 2013*; *Talavera et al., 2015*) we subjected phylogenetically corrected MI z-scores (MISTIC; [(*Simonetti et al., 2013*)]) to a significance threshold representing the top absolute z-score from all pairwise comparisons from across analyses of randomized datasets (n = 150), as previously described (*Ashenberg and Laub, 2013*. These MI calculations were conducted using MISTIC on the Teleost and Tetrapod RH1 amino acid alignments, separately, and phylogenetically corrected MI z-scores were reported for sites within a 6 Å radius of site 122 (*Table 4*).

Lastly, to further corroborate our $d_N/d_S$ analyses we investigated for evidence of correlated evolution between site 122 amino acid variation and variation at other sites within a 6 Å radius. This was done using an amino acid alignment of Teleost RH1 only; Tetrapod RH1 was not analyzed since site 122 is invariant. Consensus amino acid residues were determined for each site that fell within the 6 Å radius, where a consensus residue at a given position within a given taxa was represented as a '0', whereas a natural variant was numbered as '1'. A phylogenetic method (*Pagel, 1994*) was then used to test for correlated evolution in amino acid variation between a given site within a 6 Å radius of site 122. The Teleost species phylogeny described above was used for these analyses within the MESQUITE software package (*Maddison and Maddison, 2017*, where p-values were calculated by performing Monte Carlo tests using data from simulations (n > 1000) as previously described (*Pagel, 1994*). Significance was determined using p-values subjected to a Bonferroni-correction for multiple testing (*Table 4*).

## Ancestral reconstruction

To reconstruct the evolutionary history of sites 119, 122, 123 and 124 at the origin of both Tetrapods and the Characiphysi, we used the Vertebrate *rh1* alignment and phylogeny described above. This dataset was then used to implement codon-based marginal ancestral sequence reconstructions using the PAML 4.7 software package (*Yang, 2007*). Ancestral sequences were chosen from the best-fitting random sites model, which was M8 (*Table 7*). The likelihood-based reconstruction uses branch lengths and relative substitution rates between nucleotides, followed by empirical Bayesian reconstruction of ancestral codon states at ancestral nodes, where uncertainty is measured as

**Table 10.** Analyses of selection on Characiphysi rhodopsin (*rh1*) using PAML random sites models.

| Model | *ln*L | Parameters[1] | | | Null | *P* [df][2] | Δ AIC[§] |
|---|---|---|---|---|---|---|---|
| | | $\omega_0$/p | $\omega_1$/q | $\omega_2$/$\omega_p$ | | | |
| M0 | −10819.13 | 0.06 | - | - | N/A | - | 842.6 |
| M1a | −10586.68 | 0.03 (91%) | 1.00 (9%) | - | M0 | 0.000 [1] | 379.73 |
| M2a | −10586.68 | 0.03 (91%) | 1.00 (9%) | 9.07 (0%) | M1a | 1 [(2 | 383.7 |
| M3 | −10403.20 | 0.00 (60%) | 0.08 (29%) | 0.40 (11%) | M0 | 0.000 4) | 18.8 |
| M7 | −10401.45 | 0.17 | 1.77 | - | N/A | - | 9.27 |
| M8a | −10395.82 | 0.18 | 2.27 | 1.00 | N/A | - | 0.23 |
| M8 | −10394.82 | 0.18 | 2.22 | 1.50 | M7 | 0.000 (2) | 0* |
| | | | | | M8a | 0.136 [1] | |

*[†] $\omega$ values of each site class are shown are shown for model M0-M3 ($\omega_0$– $\omega_2$) with the proportion of each site class in parentheses. For M7 and M8, the shape parameters, p and q, which describe the beta distribution are listed instead. In addition, the $\omega$ value for the positively selected site class ($\omega_p$, with the proportion of sites in parentheses) is shown for M8.

[‡]Significant p-values (α ≤0.05) are bolded. Degrees of freedom are given in square brackets after the p-values.

 Model fits were assessed by Akaike information criterion differences to the best fitting model (asterisk).

Abbreviations—*ln*L, ln Likelihood; p, p-value; N/A, not applicable.

DOI: https://doi.org/10.7554/eLife.35957.022

posterior probabilities (*Yang, 2006*. To identify ancestral codons at the ancestral nodes (*Figure 2*), we consulted the full posterior probability distribution from the marginal reconstruction, where the character with the highest posterior probability is the best reconstruction (*Yang, 2006*. We verified the complete conservation of F119/I122/N123/S124 in Characiphysi RH1 by reference to an expanded Characiphysi RH1 amino acid alignment we assembled using a wide phylogenetic sampling of publicly available *rh1* sequences (*Supplementary file 3*).

## Rhodopsin mutagenesis, expression and spectroscopic assays

The complete coding sequence of bovine (*Bos taurus*) rhodopsin in the pJET1.2 cloning vector (ThermoFisher Scientfic), as described in a previous study was used here (*Castiglione et al., 2017*). Site-directed mutagenesis primers were designed to induce single amino acid substitutions via PCR (QuickChange II, Agilent). All sequences were verified using a 3730 DNA Analyzer (Applied Biosystems) at the Centre for Analysis of Genome Evolution and Function (CAGEF) at the University of Toronto. Wild type and mutant rhodopsin sequences were transferred to the pIRES-hrGFP II expression vector (Stratagene) for subsequent transient transfection of HEK293T cells (8 μg per 10 cm plate) using Lipofectamine 2000 (Invitrogen). HEK293T cells were obtained from David Hampson (University of Toronto), were authenticated by STR profiling (Centre for Applied Genomics, The Hospital for Sick Children) and tested negative for mycoplasma contamination. Media was changed after 24 hr, and cells were harvested 48 hr post-transfection. Cells were washed twice with harvesting buffer (PBS, 10 μg/mL aprotinin, 10 μg/mL leupeptin), and rhodopsins were regenerated for 2 hr in the dark with 5 μM 11-*cis*-retinal generously provided by Dr. Rosalie Crouch (Medical University of South Carolina). After regeneration the samples were incubated at 4°C in solubilisation buffer (50 mM Tris pH 6.8, 100 mM NaCl, 1 mM CaCl2, 1% dodecylmaltoside, 0.1 mM PMSF) for 2 hr and immunoaffinity purified overnight using the 1D4 monoclonal antibody coupled to the UltraLink Hydrazide Resin (ThermoFisher Scientific). Resin was washed three times with wash buffer 1 (50 mM Tris pH 7.0, 100 mM NaCl, 0.1% dodecylmaltoside) and twice using wash buffer 2 (50 mM sodium phosphate, 0.1% dodecylmaltoside; pH 7.0). Rhodopsins were eluted from the UltraLink resin using 5 mg/mL of a 1D4 peptide, consisting of the last nine amino acids of bovine rhodopsin (TETSQVAPA).

The UV-visible absorption spectra of purified rhodopsin samples (*Figure 4—figure supplement 1*) were recorded in the dark at 25°C using a Cary 4000 double-beam absorbance spectrophotometer (Agilent). All $\lambda_{MAX}$ values were determined by fitting dark spectra to a standard template curve for A1 visual pigments (*Govardovskii et al., 2000*. Rhodopsin samples were light-activated for 30 s using a fiber optic lamp (Dolan-Jenner), resulting in a shift in $\lambda_{MAX}$ to ~380 nm, characteristic of the biologically active metarhodopsin II intermediate (*Van Eps et al., 2017*).

Retinal release following rhodopsin photoactivation was monitored using a Cary Eclipse fluorescence spectrophotometer equipped with a Xenon flash lamp (Agilent), according to a protocol modified from previous studies (*Schafer et al., 2016*; *Farrens and Khorana, 1995*). Rhodopsin samples (0.1 – 0.2 μM) were bleached for 30 s at 20°C with a fiber optic lamp (Dolan-Jenner) using a filter to restrict wavelengths of light below 475 nm to minimize heat. Fluorescence measurements were recorded at 30 s intervals with a 2 s integration time, using an excitation wavelength of 295 nm (1.5 nm slit width) and an emission wavelength of 330 nm (10 nm slit width). There was no noticeable activation by the excitation beam prior to rhodopsin activation. This assay detected increasing fluorescence as a result of decreased quenching of intrinsic tryptophan fluorescence at W265 by the retinal chromophore (*Farrens and Khorana, 1995*, and is a reliable proxy for the tracking the decay of MII (*Schafer et al., 2016*). Data was fit to a three variable, first-order exponential equation ($y = y_0 + a(1-e^{-bx})$), and half-life values were calculated using the rate constant $b$ ($t_{1/2} = \ln2/b$). All curve fittings resulted in $r^2$ values greater than 0.95. Differences in retinal release half-life values were statistically assessed using a two-tailed $t$ test with unequal variance.

## Homology modelling of L119F/E122I/I123N/A124S Metarhodopsin II

To better evaluate the potential for natural variants at sites 119, 122, 123 and 124 to disrupt nearby structural motifs of rhodopsin, the L119F/E122I/I123N/A124S quadruple mutant structure was computationally estimated from the 3D structure of MII (PDB code: 3PQR) (*Choe et al., 2011*. A 3D structure of MII with all-*trans*-retinal bound was inferred *via* homology modelling by MODELLER (*Sali and Blundell, 1993*; *Eswar et al., 2006*133,134). Minimizing the MODELLER objective function generated 100 separate models, and the run with the lowest discrete optimized protein energy (DOPE) score was assessed (*Shen and Sali, 2006*), with reference to the next four best fitting models serving as validation of structural changes. For each estimated structure, ProCheck was used to verify the high probability of bond angle and length stereochemical conformations, as indicated by positive overall *G*-factor (*Laskowski et al., 1993*. Comparisons of each model's total energy to that expected by random chance were examined using ProSA-web (*Wiederstein and Sippl, 2007*). Images of 3D structures were generated using the PyMOL molecular graphics system, version 1.3 (Schrödinger, LLC).

## Acknowledgements

This work was supported by a National Sciences and Engineering Research Council (NSERC) Discovery grant (BSWC) and a Vision Science Research Program Scholarship (GMC). The 11-*cis*-retinal was generously provided by Rosalie Crouch (Medical University of South Carolina). We thank Alex Van Nynatten, Eduardo de A. Gutierrez, Frances E. Hauser, Nihar Bhattacharyya, and James Morrow for helpful discussions, as well as Alexandra Rui Yue and Matthew Preston for assistance formatting data.

## Additional information

### Funding

| Funder | Grant reference number | Author |
|---|---|---|
| Natural Sciences and Engineering Research Council of Canada | Discovery grant | Belinda S Chang |

The funders had no role in study design, data collection and interpretation, or the decision to submit the work for publication.

### Author contributions

Gianni M Castiglione, Conceptualization, Data curation, Formal analysis, Validation, Investigation, Visualization, Writing—original draft; Belinda SW Chang, Conceptualization, Resources, Supervision, Funding acquisition, Project administration, Writing—review and editing

## Author ORCIDs
Gianni M Castiglione (iD) http://orcid.org/0000-0002-0768-4236
Belinda SW Chang (iD) http://orcid.org/0000-0002-6525-4429

## Decision letter and Author response
Decision letter https://doi.org/10.7554/eLife.35957.028
Author response https://doi.org/10.7554/eLife.35957.029

## Additional files

### Supplementary files

• Supplementary file 1. Tetrapods and non-tetrapod Sarcopterygian outgroups with their respective rhodopsin coding-sequence (*rh1*) accession numbers.
DOI: https://doi.org/10.7554/eLife.35957.023

• Supplementary file 2. Teleost fishes and cartilaginous outgroups with their respective rhodopsin coding-sequence (*rh1*) accession numbers.
DOI: https://doi.org/10.7554/eLife.35957.024

• Supplementary file 3. Characiphysi (Siluriformes, Gymnotiformes, and Characiformes) with Cypriniforme outgroups with their respective rhodopsin coding-sequence (*rh1*) accession numbers.
DOI: https://doi.org/10.7554/eLife.35957.025

• Transparent reporting form
DOI: https://doi.org/10.7554/eLife.35957.026

All data generated or analysed during this study are included in the manuscript and supporting files.

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
