## [Decision Letter]

Thank you for submitting your article "Functional Trade-offs and Environmental Variation Determined Ancient Trajectories in the Evolution of Dim-light Vision" for consideration by *eLife*. Your article has been reviewed by three peer reviewers, including Andrei N Lupas as the Reviewing Editor and Reviewer #1, and the evaluation has been overseen by Patricia Wittkopp as the Senior Editor. The following individual involved in review of your submission has agreed to reveal their identity: Karen Carleton (Reviewer #3).

The reviewers have discussed the reviews with one another and the Reviewing Editor has drafted this decision to help you prepare a revised submission.

Summary:

This article explores residue 122 of rhodopsin (bovine sequence numbering), entirely conserved as glutamate in tetrapods, and its immediate sequence environment (the motif LxxEIA). The original observation that E122 differentiates rod and cone visual pigments was made some time ago and was supported by the impact that this residue has in increasing pigment stability, albeit at the cost of diminishing photosensitivity. More recently, it has become apparent that certain species, in particular of teleost fishes, have a range of substitutions at this site, which do not appear to entail a trade-off between stability and sensitivity (such as the motif FxxINS). The inability of tetrapods to circumvent E122, despite selection in low-light environments, is therefore puzzling. Replacement of the LxxEIA motif in bovine rhodopsin with FxxINS indeed yielded a protein with comparable stability to wild-type but with a blue-shifted spectral sensitivity, showing that the solution found in fish was in principle also available to tetrapods. However, studies of intermediate forms between the two motifs showed a discontinuous evolutionary landscape with no favorable intermediates bridging the sequence space in between. From this the authors conclude that the favorable endpoint found in some fish lineages could not be reached by tetrapods because of functional trade-offs between rhodopsin-mediated photosensitivity and photoprotection, which prevented a transition away from E122.

This is a fascinating exploration of evolutionary trajectories and the role of sequence changes in affecting rhodopsin function. The experimental analysis of the relative properties of different rhodopsin variants in the context of an evolutionary landscape provides an important advance in our understanding of rhodopsin. The following lists a number of issues which would improve the study and should be addressed.

Essential revisions:

Why was the computational analysis of residues co-evolving with position 122 not performed for the entire protein? A cut-off distance of 6 Å seems arbitrary and not supported by the well-documented observation that functional residue networks connected to specific activities are often extended and that residues in the second shell, and sometimes in the third shell, around a functional site can substantially affect its properties.

Even with the 6 Å cutoff, the decisions affecting the prioritization of given residues for experimental study are not clear. According to Table 4, positions 127 and 168 are both more clearly co-evolving with position 122 than position 123 is, yet were not analyzed, while position 123 was included. In light of the fact that N123 turned out to be necessary for the full rescue of the E122I mutation, it seems highly likely that the evolutionary landscape around position 122 is more complex than discussed in the article and shown in Figure 4.

The authors address the fitness landscape between the LxxEIA and FxxINS motifs by studying transition forms, but it seems unlikely that one evolved from the other. It seems more likely that there was an ancestral sequence which resolved itself in either of these two ways, leading to the final phenotypes. While reconstructing ancestral sequences comes with a measure of uncertainty that grows with the time elapsed, it would still be useful to have the sequence of the probable ancestor as a point of reference, as well as a discussion of this point.

In light of the work-load required to obtain further experimental data in support of the conclusions, the reviewers are not asking for further experiments, but encourage the authors to add any supporting material they have obtained already, particularly in support of the trade-off interpretation. The authors propose that tetrapods might have acquired a more stable MII state for rhodopsin as it might protect against photodamage; this is interesting, but a weakness is that animals that rely on rods would be under lower light conditions where photodamage was less likely. One might think that cones would be the photoreceptors that need more protection against photodamage. How does this fit with the known cone sequences? Are they more or less stable than rods?

Given the uncertainty in the shape of the evolutionary landscape around position 122 and the interpretation of the findings based on a limited dataset, the authors should avoid overstating their conclusions and should reflect this uncertainty in the Discussion.

[Editors' note: further revisions were requested prior to acceptance, as described below.]

Thank you for resubmitting your work entitled "Functional Trade-offs and Environmental Variation Determined Ancient Trajectories in the Evolution of Dim-light Vision" for further consideration at *eLife*. Your revised article has been reviewed by two reviewers, including Andrei Lupas as the Reviewing Editor, and the evaluation has been overseen by Patricia Wittkopp as the Senior Editor. The following individual involved in review of your submission has agreed to reveal their identity: Karen Carleton.

The manuscript has been improved but there are some remaining issues that need to be addressed before acceptance, as outlined below:

In response to the first essential point in the first decision letter ("Why was the computational analysis of residues co-evolving with position 122 not performed for the entire protein?"), you included new Figures 2B and D, and revised Table 4, to show protein-wide results for sites coevolving with position 122. This new material indeed bears out much more clearly your decision to limit the experimental analysis to residues within 6Å of position 122. It however also highlights the strong signal for positions 127 and 168, which were the subject of the second essential point ("… the decisions affecting the prioritization of given residues for experimental study are not clear."). The rationale for excluding position 127 from analysis, given in the last paragraph of the subsection “Phylogenetic identification of an intramolecular coevolutionary network”, seems somewhat speculative and is not supported by other analyses in the paper (e.g. Table 4). More importantly, this rationale does not cover position 168, which in all analyses presented appears to be strongly correlated with position 122. We would welcome a more detailed comparative discussion of all correlated sites revealed by your analyses, including position 168 and the reasons for its exclusion from experimental analysis.

We brought up the uncertainty in the evolutionary landscape around position 122 resulting from the limited dataset (such as from omission of positions 127 and 168) in the fifth essential point in the first decision letter ("… the authors should avoid overstating their conclusions and should reflect this uncertainty in the Discussion."). In response, you have added to the Discussion (subsection “Exploring inferred fitness landscapes using natural variation”, end of second paragraph), but the conclusions continue to be stated as strongly as before in all parts of the manuscript, including title and Abstract. Figure 5 (formerly Figure 4) also seems unchanged. We would welcome a more careful pruning of over-interpretations throughout the paper.

---

## [Author Response]

Essential revisions:Why was the computational analysis of residues co-evolving with position 122 not performed for the entire protein? A cut-off distance of 6 Å seems arbitrary and not supported by the well-documented observation that functional residue networks connected to specific activities are often extended and that residues in the second shell, and sometimes in the third shell, around a functional site can substantially affect its properties.

We agree that the criteria for our computational analyses were poorly communicated. We had not included a description of our initial scan of coevolution across the entire transmembrane domain (residues 53-302), where significant evidence for coevolution with site 122 occurred primarily at RH1 positions within <6 Å of E122 in the MII crystal structure. We have now added this data to the main text (subsection “Phylogenetic identification of an intramolecular coevolutionary network”) and constructed a new Figure 2B. This includes mentioning the caveat in the aforementioned subsection and Discussion (subsection “Exploring inferred fitness landscapes using natural variation”, end of second paragraph) that there may be other positions coevolving with site 122 that fall outside of this 6 Å radius. However, our analysis of coevolving sites detected within the 6 Å radius were consistent with functional data from studies of human pathogenic mutations (e.g. A164V) thought to disrupt the MII-stabilizing E122-H211 interaction (subsection “Phylogenetic identification of an intramolecular coevolutionary network”, second paragraph). This suggested that natural variation at coevolving sites within this radius could potentially compensate for the functional effects of COV at site 122. We therefore decided to focus our investigations on identifying natural compensatory mutations at sites within this 6 Å radius.

Even with the 6 Å cutoff, the decisions affecting the prioritization of given residues for experimental study are not clear. According to Table 4, positions 127 and 168 are both more clearly co-evolving with position 122 than position 123 is, yet were not analyzed, while position 123 was included. In light of the fact that N123 turned out to be necessary for the full rescue of the E122I mutation, it seems highly likely that the evolutionary landscape around position 122 is more complex than discussed in the article and shown in Figure 4.

We agree that this could have been more clear, and that these caveats should be included. We have now substantially revised the figures (Figure 2A-D; Figure 3A-B) and text to clearly communicate the rationale used to make these decisions (subsection “Phylogenetic identification of an intramolecular coevolutionary network”), which includes drawing attention to sites 127 and 168. Briefly, we used several lines of evidence to identify sites coevolving with position 122, including mutual information analyses of covariation, structural analyses, analyses of natural variation including ancestral reconstruction, and codon-based likelihood methods demonstrating evidence of assignment to site categories under increases in purifying selection.

The authors address the fitness landscape between the LxxEIA and FxxINS motifs by studying transition forms, but it seems unlikely that one evolved from the other. It seems more likely that there was an ancestral sequence which resolved itself in either of these two ways, leading to the final phenotypes. While reconstructing ancestral sequences comes with a measure of uncertainty that grows with the time elapsed, it would still be useful to have the sequence of the probable ancestor as a point of reference, as well as a discussion of this point.

We agree with the reviewers that this is a useful point of reference. We have now estimated this ancestral sequence and created a new figure to emphasize this point (Figure 3B), which is discussed in the main text and figure legend (subsection “Phylogenetic identification of an intramolecular coevolutionary network”, last paragraph).

In light of the work-load required to obtain further experimental data in support of the conclusions, the reviewers are not asking for further experiments, but encourage the authors to add any supporting material they have obtained already, particularly in support of the trade-off interpretation. The authors propose that tetrapods might have acquired a more stable MII state for rhodopsin as it might protect against photodamage; this is interesting, but a weakness is that animals that rely on rods would be under lower light conditions where photodamage was less likely. One might think that cones would be the photoreceptors that need more protection against photodamage. How does this fit with the known cone sequences? Are they more or less stable than rods?

We thank the reviewers for these very helpful comments. We have added information to the text (Introduction, third paragraph; subsection “Physiological relevance of MII stability – a proposed role in photoprotection in the eye”, first and fourth paragraphs) along with a new figure (Figure 1B) that altogether explain how evidence from physiology and biochemistry supports our trade-off interpretation, which includes specific mention of differences in rod vs. cones.

Given the uncertainty in the shape of the evolutionary landscape around position 122 and the interpretation of the findings based on a limited dataset, the authors should avoid overstating their conclusions and should reflect this uncertainty in the Discussion.

We thank the reviewers for this important feedback. We have added text to the Discussion to reflect that the limitations of our dataset, while also mentioning that the uncertainty of the site 122 evolutionary landscape likely makes the sequence-function relationship of rhodopsin highly complex (subsection “Exploring inferred fitness landscapes using natural variation”, second paragraph). We have also made attempts to draw attention to this complexity in some of our revised figures (Figure 2B-D).

[Editors' note: further revisions were requested prior to acceptance, as described below.]

The manuscript has been improved but there are some remaining issues that need to be addressed before acceptance, as outlined below:In response to the first essential point in the first decision letter ("Why was the computational analysis of residues co-evolving with position 122 not performed for the entire protein?"), you included new Figures 2B and D, and revised Table 4, to show protein-wide results for sites coevolving with position 122. This new material indeed bears out much more clearly your decision to limit the experimental analysis to residues within 6Å of position 122. It however also highlights the strong signal for positions 127 and 168, which were the subject of the second essential point ("… the decisions affecting the prioritization of given residues for experimental study are not clear."). The rationale for excluding position 127 from analysis, given in the last paragraph of the subsection “Phylogenetic identification of an intramolecular coevolutionary network”, seems somewhat speculative and is not supported by other analyses in the paper (e.g. Table 4). More importantly, this rationale does not cover position 168, which in all analyses presented appears to be strongly correlated with position 122. We would welcome a more detailed comparative discussion of all correlated sites revealed by your analyses, including position 168 and the reasons for its exclusion from experimental analysis.

We agree that a more detailed comparative discussion was warranted. We have now substantially revised and added text to clarify the integrative rationale used to make these decisions (subsection “Phylogenetic identification of an intramolecular coevolutionary network”, fifth paragraph), which now features an explicit comparative discussion of sites 127 and 168 and why they were excluded from experimental analysis.

We brought up the uncertainty in the evolutionary landscape around position 122 resulting from the limited dataset (such as from omission of positions 127 and 168) in the fifth essential point in the first decision letter ("… the authors should avoid overstating their conclusions and should reflect this uncertainty in the Discussion."). In response, you have added to the Discussion (subsection “Exploring inferred fitness landscapes using natural variation”, end of second paragraph), but the conclusions continue to be stated as strongly as before in all parts of the manuscript, including title and Abstract. Figure 5 (formerly Figure 4) also seems unchanged. We would welcome a more careful pruning of over-interpretations throughout the paper.

We thank the reviewers for providing this important feedback. In addition to the lines we previously added (subsection “Exploring inferred fitness landscapes using natural variation”, second paragraph), we have also added text to the beginning of the Discussion emphasizing the experimental limitations of our dataset (first paragraph). Importantly, we have also replaced the strong phrasing “determined” with “shaped” in the title, and have added qualifiers to the Abstract, Figure 5 legend, while also revising or removing overstated conclusions there and throughout the rest of the Discussion.